# Many-body delocalization dynamics in long Aubry-André quasi-periodic chains

Elmer V. H. Doggen[1, *] and Alexander D. Mirlin[1, 2, 3, 4]

[1]*Institut für Nanotechnologie, Karlsruhe Institute of Technology, 76021 Karlsruhe, Germany*
[2]*Institut für Theorie der Kondensierten Materie, Karlsruhe Institute of Technology, 76128 Karlsruhe, Germany*
[3]*L. D. Landau Institute for Theoretical Physics RAS, 119334 Moscow, Russia*
[4]*Petersburg Nuclear Physics Institute, 188300 St. Petersburg, Russia*

(Dated: September 17, 2019)

We study quench dynamics in an interacting spin chain with a quasi-periodic on-site field, known as the interacting Aubry-André model of many-body localization. Using the time-dependent variational principle, we assess the late-time behavior for chains up to $L = 50$. We find that the choice of periodicity $\Phi$ of the quasi-periodic field influences the dynamics. For $\Phi = (\sqrt{5} - 1)/2$ (the inverse golden ratio) and interaction $\Delta = 1$, the model most frequently considered in the literature, we obtain the critical disorder $W_c = 4.8 \pm 0.5$ in units where the non-interacting transition is at $W = 2$. At the same time, for periodicity $\Phi = \sqrt{2}/2$ we obtain a considerably higher critical value, $W_c = 7.8 \pm 0.5$. Finite-size effects on the critical disorder $W_c$ are much weaker than in the purely random case. This supports the enhancement of $W_c$ in the case of a purely random potential by rare "ergodic spots," which do not occur in the quasi-periodic case. Further, the data suggest that the decay of the antiferromagnetic order in the delocalized phase is faster than a power law.

## I. INTRODUCTION

Many-body localization (MBL) describes the localization of particles in an interacting many-body system due to the presence of disorder [1–4], which can be viewed as a generalization of Anderson localization [5] to interacting systems. This phenomenon is of interest for aiding our understanding of the general mechanisms by which ergodicity is broken in quantum systems, which is crucial for technological applications such as protecting qubits used for quantum computing against decoherence.

Many of the early works studying MBL, both theoretical [6, 7] and numerical [8–10], focused on the case where one considers a system that is Anderson-localized, adding interactions to such a system. Such systems are popularly modelled using one-dimensional (1D) disordered spin chains, such as the XXZ Heisenberg chain with a random on-site field, or the equivalent (through a Jordan-Wigner transformation) model of interacting hard-core bosons with a random on-site potential. However, some recent experiments [11–13] using ultra-cold atoms instead use a related but slightly different system, where the on-site potential is not purely random but rather quasi-periodic. The corresponding non-interacting model, known in 1D as the Aubry-André model [14], differs from the Anderson model. The transition from a localized eigenspectrum to a delocalized one occurs not at infinitesimal disorder strength, but at a finite value of $W = 2$ (in units used in this work). Nevertheless, an MBL transition from a delocalized to a localized eigenspectrum is still expected to occur in the interacting case at a critical disorder $W_c > 2$ [15].

In purely random systems, numerical studies find the appearance of power laws in transport properties, which are attributed to Griffiths effects [16, 17] associated with rare events. In disordered interacting systems two interrelated types of rare events have been identified: (i) rare thermal regions that serve to delocalize an otherwise localized phase, and (ii) rare strongly disordered regions that act as bottlenecks for transport [18].

A rare thermal region, also termed "ergodic spot", consists of a region of the disordered potential that is anomalously weak in comparison with its typical values in the system. To make this more concrete, consider an uncorrelated random potential $\phi_i$ with values $\forall i$ taken from a uniform distribution $\in [-W/2, W/2]$. There is always a finite probability that over some stretch of length $l$ sites the potential takes only values whose site-to-site magnitude variations (i.e. the difference between the maximum and minimum values of the potential) are limited by $W'$ where $W' < W$. For small $W'$ and large $l$ such regions are exponentially unlikely, but they may still be of importance in the thermodynamic limit, as was rigorously shown in the case of the low-energy properties of disordered bosons in one dimension [19]. Recent theoretical work based on renormalization group approaches [20, 21] suggests that rare ergodic spots lead to "avalanches" enhancing many-body delocalization, as also supported by numerical results [22]. This was argued to shift the position of the MBL transition in favor of the delocalized phase and also to control scaling near the transition.

With an analogous reasoning, one can show that anomalously *strongly* disordered regions must exist, characterized by an effective disorder that is larger than that associated with the typical fluctuations of $\phi_i$. Again, the probability of a strongly disordered region of length $l$ is exponentially suppressed with $l$. At the same time such a region serves as an effective tunnel barrier with transparency that is also exponentially small in $l$. An interplay of two such exponentially small quantities is characteristic for Griffiths effects, leading to slow (subdiffusive power-law) transport.

---

* Corresponding author: elmer.doggen@kit.edu

Experimentally, power laws describing particle transport are also found in quasi-periodic disordered systems [12]. This is rather unexpected, since rare events should be absent in a deterministic potential, such as a quasi-periodic potential. This is because in a quasi-periodic system the values $\phi_i$ of the potential are not uncorrelated but rather taken from a deterministic formula. Therefore, extended regions with anomalously small or large disorder are not possible. Hence, the reasoning outlined above that is based on the emergence of rare regions does not apply, and we ought to find differences in the dynamics between disordered and quasi-periodic systems if such regions are indeed important in the disordered case, which is one of the key objectives of this work.

In a recent work, Khemani *et al.* studied the difference between the two types of disorder [23]. They describe two distinct universality classes, and argue that previous numerical studies for purely random disorder, focusing primarily on exact diagonalization studies of small systems, are strongly affected by finite-size effects. The authors attribute the violation of Harris-Chayes bounds [24] in numerical studies of small systems $L \leq 20$ to such effects. Indeed, our recent numerical results confirm that exact diagonalization studies of purely random models are tainted by strong finite-size effects and substantially underestimate the critical disorder strength [25].

One should wonder whether the appearance of power laws in quasi-periodic systems is merely an artefact of studying insufficiently large systems and long times. Numerically, this is difficult because on the one hand, exact-diagonalization studies can easily probe long times accurately, but the small systems thus accessible "feel" the effects of boundary conditions more strongly at late times. In one such exact diagonalization study, analysis of the return probability in quasi-periodic systems shows robust power-law tails for systems of 16 lattice sites [26]. On the other hand, a recent study using a self-consistent Hartree-Fock approximation suggests that the power laws seen in the quasi-periodic case are only transient, whereas they are robust in the purely random case [27] at least up to times (in units of lattice hopping) $\mathcal{O}(10^4)$. However, the Hartree-Fock approximation is *a priori* uncontrolled, and it is not clear how well it captures the exact dynamics.

The aim of the present work is to investigate the differences between purely random and quasi-periodic disorder at system sizes inaccessible to exact diagonalization, using the newly developed numerical technique of the time-dependent variational principle (TDVP) as applied to matrix product states (MPS). Using this technique, we find that finite-size effects in this system are substantially smaller than for purely random disorder in the sense that the critical disorder does not appear to have a strong dependence on system size. Furthermore, we find that the choice of the specific quasi-periodic potential substantially impacts transport properties as well as the value of the critical disorder $W_c$, which is explained by the influence of periodicity on the properties of the potential and thus of the corresponding single-particle states. We use the TDVP approach to locate the position of the transition for the most standard choice of parameters (periodicity $\Phi = (\sqrt{5} - 1)/2$ and interaction $\Delta = 1$, see definitions below), as well as for the choice $\Phi = \sqrt{2}/2$. We also find possible indications of the breakdown of power-law behavior in larger systems: the decay becomes faster than a power law with increasing time. Our findings are consistent with the aforementioned predictions of Khemani *et al.* concerning two different universality classes for the MBL transitions with random and quasi-periodic potentials [23] (see also the related study [28]).

Several MPS-based methods have been applied to the problem of MBL. "Traditional" MPS methods such as the time-dependent density matrix renormalization group (t-DMRG) [29, 30] and time-evolving block decimation (TEBD) [31] can be used, but are restricted to the MBL side of the transition or very short times [8, 32–34] due to the growth of the entropy of entanglement and non-conservation of energy. Žnidarič *et al.* pioneered an approach [35, 36] based on DMRG that, on the other hand, is particularly suitable for the ergodic regime, but does not work efficiently closer to the localized regime. TDVP achieves a middle ground, allowing for the access of intermediate times in the weakly ergodic regime close to the transition, as well as long times in the localized regime (like other MPS methods), thus complementing the earlier approaches.

## II. MODEL AND METHOD

### A. Quasi-periodic Heisenberg XXZ chain

We consider the Heisenberg XXZ chain with a quasi-periodic field, also known as the interacting Aubry-André model, on a lattice of length $L$ with open boundary conditions, as described by the Hamiltonian

$$\mathcal{H} = \sum_{i=1}^{L} \left[ \frac{J}{2}\left( S_i^+ S_{i+1}^- + S_i^- S_{i+1}^+ \right) + \Delta S_i^z S_{i+1}^z + \phi_i S_i^z \right],$$
(1)

where $\phi_i = (W/2)\cos(2\pi\Phi i + \phi_0)$ represents the quasi-periodic field with strength $W$, $\Phi$ is a parameter describing the periodicity of the underlying potential and $\phi_0 \in [0, 2\pi)$ is a random phase taken from a uniform distribution. If $\Phi$ is chosen to be irrational, the period of the potential is infinite. In this work we choose $\Phi = (\sqrt{5} - 1)/2$, the inverse golden ratio, unless noted otherwise. The $S$-operators represent standard Pauli matrices, so that the case $\Delta = 1$ corresponds to the isotropic Heisenberg chain. This problem can be mapped to a particle representation using a Jordan-Wigner transformation, wherein $J$ is the hopping element for the single-particle transport to adjacent sites, and $\Delta$ represents the inter-particle interaction. In the case $\Delta = 0$, Eq. (1) maps to the Aubry-André model of non-interacting particles in a quasi-periodic field [14], which exhibits a transition from a fully ergodic (delocalized) eigenspectrum

below a critical disorder $W/J = 2$ to a fully localized spectrum above it. In the following, we set $J \equiv 1$ as the unit of energy, and $\hbar \equiv 1$.

A purely random disordered model differs from the quasi-periodic model (1) as it has elements $\phi_i$ which are uncorrelated from site to site. A quantitative comparison between the two cases requires a non-uniform distribution $\phi_i = (W/2)\cos(\varphi_i)$, with a site-dependent phase $\varphi_i \in [0, 2\pi)$, cf. Ref. [23]. However, as we show below, the properties of the MBL transition also depend quantitatively on $\Phi$, further complicating comparisons between the two types of disorder.

We numerically simulate the dynamics of an initial anti-ferromagnetic state:

$$|\psi\rangle(t = 0) = \{\uparrow, \downarrow, \ldots, \uparrow, \downarrow\}, \qquad (2)$$

and compute the imbalance $\mathcal{I}$ as a function of time, which quantifies how much of the initial anti-ferromagnetic order remains at a certain time $t$. It is defined as:

$$\mathcal{I}(t) = \frac{1}{L}\sum_{i=1}^{L}(-1)^i \langle S_i^z(t)\rangle. \qquad (3)$$

It is easy to verify that the initial value of the imbalance $\mathcal{I}(t = 0) = 1$. For an ergodic system, the long-time time-average of $\mathcal{I}(t)$ vanishes in the thermodynamic limit. The initial state (2) is appealing for studies of thermalization, because it is typically the fastest-thermalizing initial state. Furthermore, for this choice of the initial state the imbalance (3) directly probes density-density correlations [37] while only requiring knowledge of spin densities, simplifying both numerical analysis and experimental probing of MBL. In the case of purely random disorder, the imbalance decays according to a power law: $\mathcal{I}(t) \propto t^{-\beta}$ [17, 25], where $\beta$ is a disorder-dependent power law. A possible criterion for pinpointing the MBL transition is the vanishing of $\beta$, indicating saturation of the imbalance. If such a saturation can be extrapolated to $t \to \infty$ the system is localized.

### B. Time-dependent variational principle

To compute the time dynamics, we use the time-dependent variational principle (TDVP). We apply the TDVP to matrix product states (MPS) [38, 39], a type of tensor network that functions as a variational ansatz within the subspace of weakly entangled states [40]. Our implementation is identical to the one used in our previous work [25]. We use a numerical time-step of $\delta t = 0.1$ or in some cases $\delta t = 0.2$; varying the time-step somewhat did not change our results meaningfully. In the Appendix we provide further technical details on the numerical implementation. Implicitly, time evolution due to the TDVP is described by:

$$\frac{\mathrm{d}|\psi\rangle}{\mathrm{d}t} = -\mathrm{i}\mathcal{P}_{\mathrm{MPS}}\mathcal{H}|\psi\rangle, \qquad (4)$$

where $\mathcal{P}_{\mathrm{MPS}}$ projects the time evolution back onto the variational manifold. In our implementation, we start with a product state described by an MPS with bond dimension of 1, and increase the bond dimension in a modest number of time steps to a desired value $\chi$, keeping the bond dimension fixed during the further time evolution. The number of variational parameters in the matrix product state ansatz scales polynomially in $\chi$ and in the system size $L$, as opposed to the full many-body wave function, whose complexity scales exponentially in $L$. A beneficial property of the TDVP is that time evolution with such a fixed bond dimension conserves globally conserved quantities such as the energy, as opposed to older MPS-based methods, such as time-evolving block decimation [31]. Recently, the performance of the TDVP has been studied in several works [25, 41–43]; these findings suggest that the TDVP is particularly suited to studying dynamics of weakly entangled states close to the MBL transition.

### III. RESULTS

We focus on the case $L = 50$ and $\Delta = 1$, unless stated otherwise, and consider dynamics up to $t = 300$. Using the TDVP we numerically compute the dynamics and track the behavior of spin density $\langle S_i^z \rangle$ at the different sites $i \in [1, L]$ for $\mathcal{O}(500)$ different "realizations of disorder," i.e. different random values of $\phi_0$. For the sake of comparison, we also consider the case $L = 16$ where we compute the dynamics exactly using the TDVP with unrestricted bond dimension. We opt to consider only the spin density here and not, e.g., the entropy of entanglement, since the former is readily measured in the experiment while the latter is difficult to probe experimentally, despite recent advances [44].

### A. Imbalance

#### 1. Dynamics of the imbalance

First, consider the imbalance (3). We compute the dynamics using the TDVP. In Fig. 1 we show dynamics for various choices of $W$ on both sides of the transition. For relatively strong disorder, $W > 5$, we consider a bond dimension of $\chi = 32$, whereas for weaker disorder we use $\chi = 32$ and $\chi = 64$. (In Sec. III B below, we will use $\chi$ up to 128 for a specific choice of periodicity and disorder, to explore more accurately the imbalance decay in the delocalized phase.) A reduction in bond dimension will typically lead to enhanced delocalization [25], implying that if we find saturation at a certain $\{W, \chi\}$ this saturation likely persists in the limit $\chi \to \infty$.

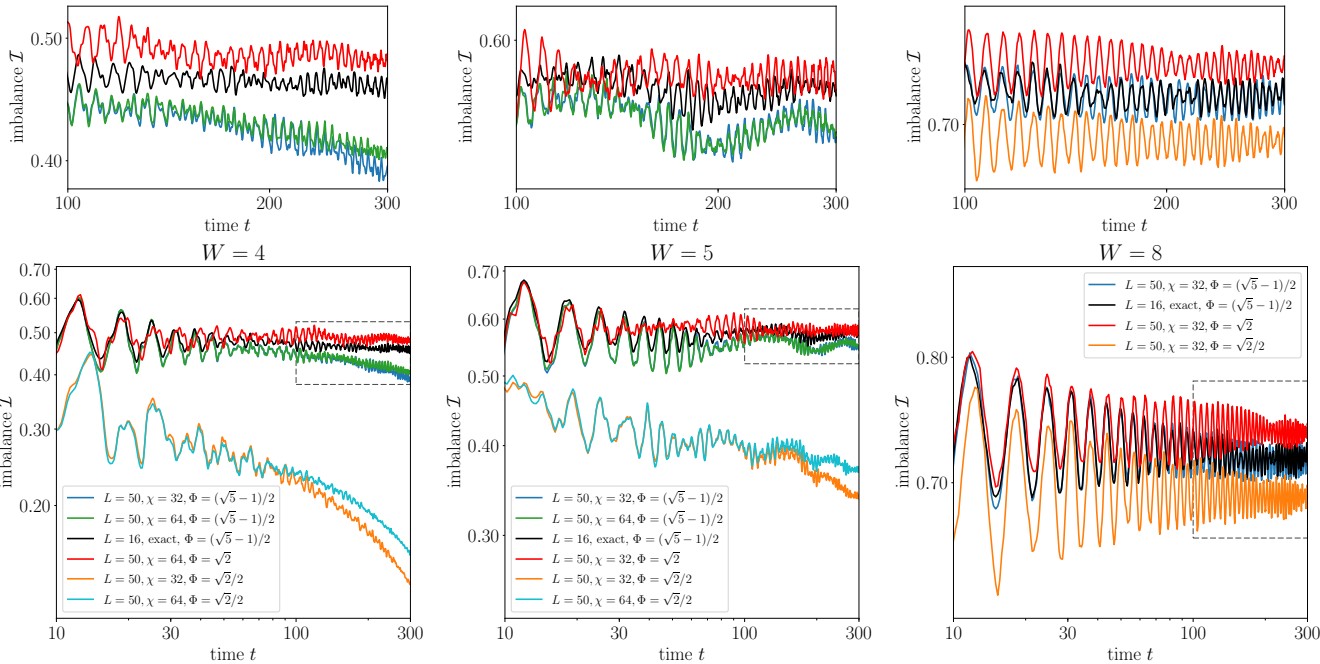

FIG. 1. Time evolution of the imbalance (3) for various choices of disorder strength $W$, periodicity $\Phi$, and bond dimension $\chi$, averaged over disorder realizations (i.e. different random phases $\phi_0$). Coloured solid lines indicate the results for lattice size $L = 50$ and $\Phi = (\sqrt{5} - 1)/2$, the dotted black line shows numerically exact results for $L = 16$ at the same value of $\Phi$. Dashed lines show results for $L = 50$ using different choices for the parameter $\Phi = \{\sqrt{2}, \sqrt{2}/2\}$. Results shown for different $\chi$ use independent disorder realizations. Top panels show zoomed regions indicated by the rectangles in the lower panels. Note the log-log scales and differing $y$-axes. Convergence with $\chi$ occurs at different times for different $W$ as well as different $\Phi$ for the same $W$ (see text).

### 2. Dependence on the periodicity $\Phi$

Strikingly, the dynamics is substantially altered by changing the periodicity parameter $\Phi$. For $W = 5$, the imbalance appears to saturate for the choices $\Phi = (\sqrt{5} - 1)/2$ and $\Phi = \sqrt{2}$, but decays for $\Phi = \sqrt{2}/2$, with substantially different values of the disorder-averaged imbalance as a function of time, suggesting that the critical disorder $W_c$ varies with $\Phi$. We also observe convergence with $\chi$ over the full time window in the cases $\Phi = \sqrt{2}$ and $\Phi = (\sqrt{5} - 1)/2$, but not for $\Phi = \sqrt{2}/2$, which indicates that the growth of the entropy of entanglement also significantly depends on $\Phi$. We have verified that this is due to the interplay between interactions and disorder by comparing to the non-interacting case $\Delta = 0$ (see left panel of Fig. 2), in which case the imbalance saturates for the choices of $\Phi$ considered here. The dependence of the imbalance decay on $\Phi$ is also clearly seen for $W = 4$. Indeed, we observe at most a very weak decay for the choice $\Phi = \sqrt{2}$, a significant decay for the inverse golden ratio $\Phi = (\sqrt{5} - 1)/2$, and a much stronger decay for $\Phi = \sqrt{2}/2$. In the latter case, numerical convergence with $\chi$ breaks down already before $t = 100$. Hence, the convergence time is a function of both $\chi$ and $\Phi$, which is related to the critical disorder itself being a function of $\Phi$.

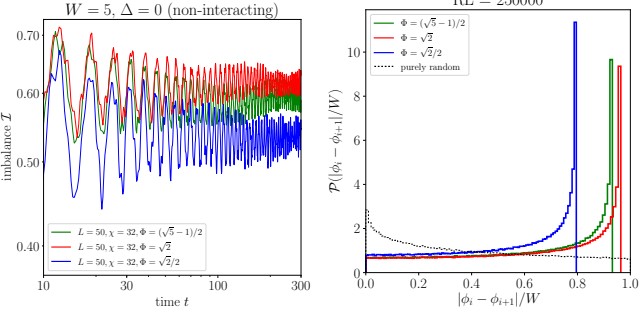

FIG. 2. **Left**: evolution of the average imbalance in the non-interacting case $\Delta = 0$ for disorder $W = 5$, computed using the TDVP. The imbalance saturates at different values for different choices of the periodicity $\Phi$. **Right**: histograms of the distribution of the differences between consecutive values of the disordered on-site potential, for various choices of $\Phi$. For comparison, also the purely random case is shown. The number of realizations is $R = 5000$, and system size $L = 50$.

The dependence on $\Phi$ can be explained in the following way. Consider the consecutive values of the disordered potential at sites $\phi_i$ and the next site $\phi_{i+1}$. We plot the distribution $\mathcal{P}$ of the absolute differences $|\phi_i - \phi_{i+1}|$ normalized by the disorder strength $W$ in the right panel of Fig. 2. For each value of $\Phi$, the support of the dis-

tribution is limited from above by a certain value $< 1$ which is different for different values of $\Phi$. The corresponding peak in the distribution is similar in origin to the van Hove singularities in the density of states [45, 46]. Hence, there is an effective disorder strength $\tilde{W}(\Phi) = W|\sin(\pi\Phi)| < W$ which characterizes the potential variation between the neighboring sites. (The analytical expression for this effective disorder is derived in Ref. [47]). Since we consider quite strong disorder, for which the localization length of the non-interacting problem is of the order of lattice spacing, this strongly influences the spatial extent of localized states. This is supported by the data for the imbalance in non-interacting problem, see left panel of of Fig. 2. Indeed, the value at which the imbalance saturates depends on $\Phi$, consistent with the "effective disorder strength" (width of the distribution) in the right panel. A quantitative dependence of quench dynamics on the choice of $\Phi$ was also found for an extended version (including next-nearest neighbor hopping terms) of the non-interacting, extended Aubry-André model [48], and $\Phi$ was recently shown to influence the appearance of mobility edges in the single-particle problem [49].

Comparing Figs. 1 and 2, we see that the dependence of the degree of localization in the single-particle problem on $\Phi$ is inherited by the many-body problem. In particular, the model with $\Phi = \sqrt{2}/2$, which has the weakest "effective disorder" and thus larger spreading of single-particle localized states out of the three considered values of $\Phi$, shows the most efficient many-body delocalization. Similarly, the case $\Phi = \sqrt{2}$, which yields the the strongest "effective disorder" and thus the smallest spreading of single-particle localized states, is the most resistant with respect to many-body delocalization.

It is worth emphasizing that the meaning of the curves in the right panel of Fig. 2 for the quasi-periodic and purely random cases is somewhat different. Specifically, for the random case, the potentials on different sites are uncorrelated. Thus, the distribution of the potential difference is independent of the distance between the sites and is just a different representation of the distribution of the random values $\phi_i$. On the other hand, for the quasi-periodic case, the shown curves apply to the distribution for neighboring sites only and, e.g., the distribution $\mathcal{P}(|\phi_i - \phi_{i+2}|)$ looks differently (not shown). This is an implication of correlations between the potential values in the quasi-periodic case and is tied to the dramatic difference with respect to the possibility of rare events. In the purely random case, arbitrary potentials bounded by $W$ are possible. As a result, for any $W' \ll W$ and any length $l$, there is a finite probability (exponentially decaying with $l$) of a weakly disordered region of length $l$ with variations (in terms of the difference between the maximum and minimum value of the disordered potential) bounded by $|\phi_{i,\max} - \phi_{i,\min}| < W'$. At the same time, in the quasiperiodic case, such a probability is strictly zero because of the strong correlations between the values of the potential. As a con-

crete illustration, consider the quasi-periodic potential with $\Phi = (\sqrt{5} - 1)/2$ and $\phi_0 = -\pi/2$. The potential at site $i = 0$ is then 0, and the neighboring sites have values $\pm(W/2)\sin(2\pi\Phi) \approx \mp0.68(W/2)$. Only for certain values of $i$ is it possible to have two nearest-neighbor sites (say, $i$ and $i+1$) with small differences in the potential values. However, also in this case the consecutive sites ($i-1$, $i+2$, etc.) will have very different potential values of order $W$. Hence, no extended "thermal" regions are possible for the quasi-periodic potential, in contrast to the purely random case.

### 3. Critical disorder $W_c$

In the case $\Phi = (\sqrt{5} - 1)/2$, the data shown in Fig. 1 exhibit a saturation of the imbalance at $W = 5$ and its decay at $W = 4$ (for the system size $L = 50$). To obtain a more accurate estimate for the critical disorder $W_c$, we have performed the analysis of the imbalance decay for various values of $W$ with a step 0.5, see Fig. 3. We find a substantial decay for $W \leq 4.5$ and staruration for $W \geq 5$, from which we infer an estimate for the critical disorder $W_c = 4.8 \pm 0.5$. To determine $W_c$, we fitted the imbalance $\mathcal{I}(t)$ to an effective power law $t^{-\gamma}$ in a time window $t \in [50, 180]$. When $\gamma$ vanishes (i.e., becomes indistinguishable from zero within the error bars), we conclude that the system is localized.

The following disclaimer is in order here. Our approach is numerical and therefore deals with a finite time range. So, rigorously speaking, we cannot exclude that even at $W$ larger than our identified $W_c$ there is an extremely slow decay that would eventually delocalize the system at $t \to \infty$. From this point of view, our estimate for $W_c$ should be understood as a lower bound, given the assumption that substantial decay over the considered time window is not followed by saturation at much later times. In the latter case, we would expect to find a *decrease* in the obtained power-law coefficient for later time windows, which is not observed (see Sec. III B).

It should be emphasized that, for the determination of saturation, the specifics of the fitting procedure are not too important since our goal here is to characterize the decay by a single parameter ($\gamma$ in the case of our fit) and to determine the value $W_c$ at which this parameter vanishes, thus indicating saturation. The decay law is reasonably close to a power law in the available time interval, which is why we choose an effective power law for the fitting. In fact, as we discuss in Sec. III B, the data indicate an acceleration of decay in comparison with a power law; however, this is not essential for the determination of $W_c$. In fact, one could also use, e.g., an exponential fit that would also show imbalance saturation at $W > W_c$.

It is also worth pointing out that, for purely random disorder, a similar analysis [25], when applied to small systems, yields results for $W_c$ that are in good agreement with those obtained from exact diagonalization [10]. We

will show below that this agreement also holds in the quasi-periodic case.

A similar analysis for $\Phi = \sqrt{2}/2$ yields a considerably higher value $W_c = 7.8 \pm 0.5$. While we did not perform such a detailed analysis for $\Phi = \sqrt{2}$, the stronger localization implies a critical disorder somewhat smaller than in the case $\Phi = (\sqrt{5} - 1)/2$. Note that these results are substantially lower than the estimate $W_c \simeq 11$ (in units of this paper) obtained for purely random disorder in large chains using the same method [25]. The distributions of on-site potential are somewhat different in both models; also, in the quasi-periodic case, we find an elevated probability for finding relatively large values of $|\phi_i - \phi_{i+1}|$ (see the peaks in Fig. 2). This may result in a somewhat smaller $W_c$. However, this cannot fully account for a large difference in $W_c$. We speculate that a considerably larger value of $W_c$ in the random problem is related to delocalizing effect of rare "ergodic spots", which are also responsible for stronger finite-size effects on the value of $W_c$, see a discussion in Sec. IV.

Clearly, the critical disorder is expected to depend on the strength of interactions $\Delta$. This dependence is, however, quite weak around the "optimal" (from the point of view of delocalizing effect) interaction $\Delta \sim 1$. We thus focus on $\Delta = 1$ in this work.

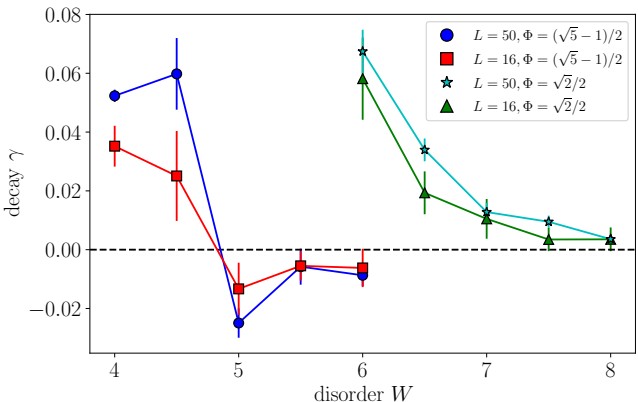

FIG. 3. Saturation of the imbalance as quantified using a power-law fit to $\mathcal{I} \propto t^{-\gamma}$ in the window $t \in [50, 180]$, for the choices $\Phi = (\sqrt{5} - 1)/2$ and $\Phi = \sqrt{2}/2$. From the saturation of the imbalance we infer estimates for the critical disorder $W_c = 4.8 \pm 0.5$ and $W_c = 7.8 \pm 0.5$ for the two choices of $\Phi$ respectively. Results for $L = 16$ are numerically exact, results for $L = 50$ use a bond dimension $\chi = 64$, except for $W = \{4, 4.5\}$ where $\chi = 128$ was used to ensure numerical convergence. Error bars in the figure are $2\sigma$-intervals estimated through a bootstrapping procedure. Lines are a guide to the eye.

We find that for both choices of $\Phi$, the value of $W$ where saturation of the imbalance is observed is very weakly dependent on system size. This behavior is in stark contrast to the purely random case [25], where saturation of the imbalance is observed to have a strong system size dependence. For disorder much stronger than

$W_c$, e.g. $W = 8$ for $\Phi = (\sqrt{5} - 1)/2$ (see Fig. 1), there is essentially no influence of the system size visible. This indicates that the localization length in this case is very small, on the order of a single lattice site. The value to which the imbalance saturates in the considered time window does depend somewhat on $\Phi$. Interestingly, on the localized side of the transition, we also find slow oscillations that do not disappear upon disorder averaging. This phenomenon is absent in the purely random case, and we attribute it to long-range correlations in the quasi-periodic potential. This is confirmed by comparing to the case where the quasi-periodic period $\Phi$ is not the inverse golden ratio, but different irrational numbers $\Phi = \{\sqrt{2}, \sqrt{2}/2\}$ (see the dashed lines in Fig. 1). We further confirm that this is not merely a finite-size effect by comparing to the case $L = 40, \Phi = (\sqrt{5} - 1)/2$ (not shown), which is quantitatively very similar to the case $L = 50$, exhibiting the same oscillations.

For weaker disorder $W \leq 4.5$ and $\Phi = (\sqrt{5} - 1)/2$, we clearly observe a decaying imbalance which is a manifestation of delocalization. In this case, we also find an increase in the rate of delocalization when the system size is increased from $L = 16$ to $L = 50$. As is seen in the figure (see also Fig. 7), for times $t \gtrsim 100$ we start to see a dependence on bond dimension, indicating that for the case $W = 4$ larger bond dimensions are required for convergence. Note that the results for different values of $\chi$ are obtained using independent realizations of the quasi-periodic fields, so that convergence is checked simultaneously with the number of realizations and the bond dimension $\chi$.

To summarize, for both system sizes $L = 16$ and $L = 50$, we find a decay of the imbalance without saturation at $W \leq 4.5$ and a saturation for $W \geq 5$. This yields the result for the critical disorder $W_c = 4.8 \pm 0.5$, with weak dependence on the system size. Qualitatively, the behavior is the same for the choice $\Phi = \sqrt{2}/2$ except that the critical disorder is at a considerably higher $W_c = 7.8 \pm 0.5$. The ratio of "effective" disorder values (see above) in the two cases is $\tilde{W}[\Phi = (\sqrt{5} - 1)/2]/\tilde{W}[\Phi = \sqrt{2}/2] \approx 1.17$, which is substantially smaller than the ratio of estimates of the critical disorder in both cases. Hence, the critical disorder cannot be expressed in a universal way in terms of $\tilde{W}$.

### 4. Critical disorder: comparisons to existing literature

Several recent papers considered, by using exact diagonalization and, in some cases, other approaches, the MBL transition in spin chains with quasi-periodic disorder or equivalent models of hard-core bosons or spinless fermions. Here we summarize some previous estimates for the critical disorder and compare them to our result $W_c = 4.8 \pm 0.5$ for the commonly used choice $\Phi = (\sqrt{5} - 1)/2$. To simplify comparison, we translate notations of other works to our conventions and notations for interaction ($\Delta$) and disorder ($W$).

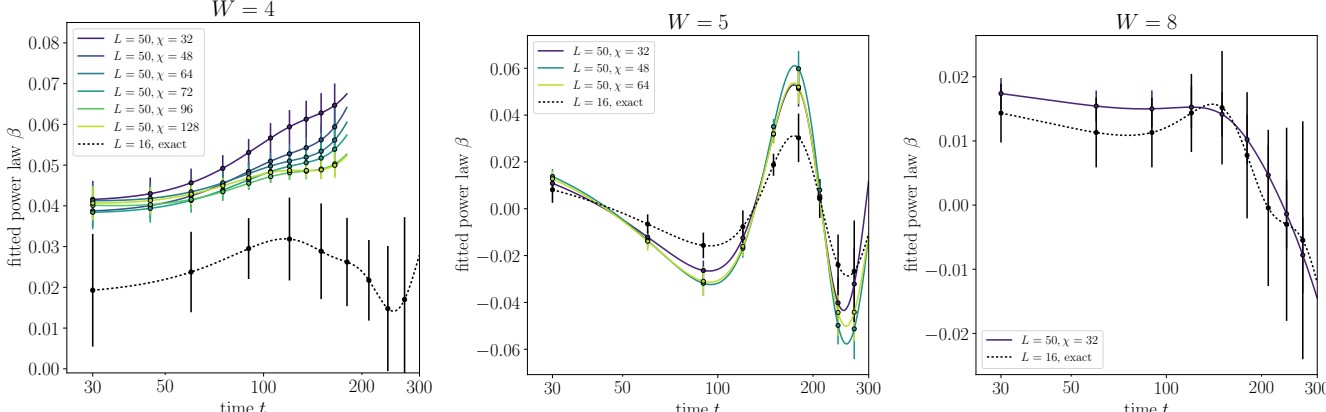

FIG. 4. Flowing power-law exponent $\beta(t)$ characterizing the decay of imbalance (3) according to the weight (5) for various values of the disorder strength $W$. Oscillations in the imbalance shown in the middle panel of Fig. 1 lead to oscillations in the time-dependent power-law coefficient $\beta$ for $W = 5$. For $W = 4, L = 50$, only times $t \leq 180$ were considered, and exceptionally for this paper also the values $\chi = \{72, 96, 128\}$, using a doubled time-step $\delta t = 0.2$ for the latter two choices. Error bars indicate $2\sigma$-intervals estimated using a bootstrapping procedure.

- In a seminal early work, Iyer *et al.* [15] applied exact diagonalization to a spin chain for various choices of the interaction. For a choice of the interaction comparable to our $\Delta = 1$, they find a rough estimate $2 \lesssim W_c \lesssim 5$ using various methods of locating the transition (see their Table II).

- Bera *et al.* [50] used exact diagonalization to study the eigenvalues of the one-body density matrix and obtained $W_c \approx 5$ for $\Delta = 1$ (see their Fig. 19).

- Bar Lev *et al.* [34] studied primarily the ergodic regime using t-DMRG and the functional renormalization group. They have also studied the level statistics via exact diagonalization and obtained the approximate phase diagram presented in their Fig. 1. This diagram suggests $W_c \approx 3$ for $\Delta = 1$; however, the authors argue that the actual value may be somewhat larger.

- Setiawan *et al.* [26] applied exact diagonalization and found, for a considerably weaker interaction $\Delta = 1/4$, an estimate $W_c = 3$ (see their Sec. II).

- Lee *et al.* [51] used exact diagonalization and apply both a finite-size scaling study of the entanglement entropy (see their Fig. 1) and the analysis of the imbalance dynamics (see their Fig. 5). For $\Delta = 1$ they find $W_c \approx 3.7$ using the former and $W_c \approx 5$ using the latter approach. However, they used $\Phi = \sqrt{2}$, for which our results indicate that the critical disorder should be somewhat lower than for $\Phi = (\sqrt{5} - 1)/2$.

- Khemani *et al.* [23] studied both purely random and quasi-periodic disorder in an *extended* spin chain model with additional next-nearest-neighbor hopping terms and found $W_c \approx 8.5$ for the quasi-periodic case (see their Fig. 4). The extended hopping term aids delocalization, so one should expect an increase in $W_c$ due to this.

- Weidinger *et al.* [27] applied a Hartree-Fock approximation to large chains up to $L = 192$ for random and quasi-periodic models with interaction $\Delta = 1/2$. For the quasi-periodic case, their Fig. 1 indicates delocalization for $W = 3$ and localization for $W = 7$, implying a localization threshold $W_c$ between these values.

Overall, previous estimates are in fair agreement with our result, underscoring that exact-diagonalization studies may perform rather well for determination of $W_c$ in quasi-periodic systems, in consistency with our conclusion on relative weakness of finite-size effects on $W_c$, see Sec. III A 3. This should be contrasted to the purely random case where exact-diagonalization studies substantially underestimate $W_c$ [25].

## B. Decay of the imbalance: power-law behavior?

A question of recent interest is whether the long-time decay of the imbalance in the ergodic regime is described by a power law or not [37], and whether there is a difference in this regard between purely random disorder and quasi-periodic disorder, as predicted numerically in Ref. [27]. To analyze the decay of the imbalance more closely, we characterize it by a (in general, time-dependent) power-law exponent $\beta(t)$. Note that we are not assuming power-law behavior in this procedure, since $\beta(t)$ varies generically with time. If we obtain an essentially constant $\beta(t)$, this would establish *a posteriori* a power-law decay of the imbalance. At the same time,

substantial deviation of $\beta(t)$ from a constant would imply that true decay is not of power-law character. In the following, and in the remainder of the paper, we consider $\Phi = (\sqrt{5} - 1)/2$.

Since we cannot probe time scales over many decades, we use the following procedure to investigate the time dependence of $\beta$. First, we consider the window of times $\tau \in [30, 300]$ ($\tau \in [30, 180]$ for $W = 4$). Then, we perform a power-law fit to $\mathcal{I}(\tau; t) \propto \tau^{-\beta(t)}$, where data points are weighted according to a Gaussian (the normalization is irrelevant here as only the relative weight influences the fitting procedure):

$$\exp\left[\frac{-(t - \tau)^2}{2\sigma^2}\right].\qquad(5)$$

We repeat this procedure for all $t$ and compute fits to $\mathcal{I}(\tau; t)$, yielding $\beta(t)$. The benefit of this procedure over using, e.g., shifting time windows to perform fits, is that it results in a smooth behavior of $\beta(t)$. The standard deviation is chosen as sufficiently large, $\sigma = 50$, such that persistent, fast oscillations due to the initial condition are washed out. Such oscillations, that do not vanish upon disorder averaging, were previously found in Ref. [34] in the dynamics of the mean-square displacement; an observation reproduced in our results (see Fig. 1). In the case of power law-behavior we expect $\beta(t)$ to be approximately constant in time, whereas an increase of $\beta$ as a function of time indicates stronger than power-law decay [27]. In a previous work on a purely random case [25], we found an essentially time-independent $\beta$, i.e., a power-law decay, for the times $t \leq 100$.

The time dependence of the flowing exponent $\beta(t)$ is shown in Fig. 4 for various choices of $W$. For $W = 4$ (left panel), i.e., on the ergodic side but fairly close to the MBL transition, we find that $\beta(t)$ for a long chain ($L = 50$) shows an increase with time, implying that the decay is faster than power law. The difficulty in numerically characterizing this increase in $\beta(t)$ is that one needs larger $\chi$ (and thus much larger computation time) to get convergence at longer times. In this calculation, we have used the bond dimension up to $\chi = 128$, which has allowed us to reach convergence up $t \approx 180$. Indeed, the results for $\chi = 96$ and $\chi = 128$ are essentially identical for such $t$. In this time window, we observe a modest increase of $\beta$, with a tendency to a further increase. Our data allow us to conjecture that the increase of $\beta(t)$ with accelerate with increasing time $t$ (consistent with Ref. [27]); verifying this in a controllable way remains a challenge for future research.

For a small system $L = 16$, the power law remains a good fit over the entire time window, as evidenced by the fact that the fitted coefficient remains constant, $\beta(t) \simeq 0.025$, to a good approximation. This is consistent with exact-diagonalization study of Ref. [26] that did not observe deviations from power-law decay. It also worth mentioning that for $L = 50$ but short times, $t \lesssim 50$, the behavior is essentially indistinguishable from a pure power law, in agreement with experimental observations and the t-DMRG results of Ref. [34] for $W = 3$.

For stronger disorder, $W \geq 5$, i.e., in the localized regime, $\beta(t)$ remains very small and shows oscillations around zero. These oscillations originate from oscillations of the imbalance in the localized regime, see Fig. 1 and the discussion in Sec. III A.

## C. Distribution of spin densities

Another quantity that can be directly probed experimentally is the distribution of spin densities $\mathcal{P}(\langle S^z \rangle)$, where we consider the normalized distribution over all sites $i$ and disorder realizations. We focus again on the case $\Phi = (\sqrt{5} - 1)/2$. A convenient feature of $\mathcal{P}(\langle S^z \rangle)$ is that it can be naturally defined also for random initial conditions where $\forall i : \langle S_i^z \rangle(t = 0) = \pm 1$, in addition to the initial Néel state used in this work. In the eigenstates of small, purely random systems [52], the study of a similar quantity reveals a bimodal distribution on the localized side of the MBL transition. A comparable picture emerges from the study of dynamics, with a bimodal distribution sharply peaked at $-1$ and 1. This is a signature of the memory of the initial state, which has a distribution consisting of two $\delta$-peaks at $-1$ and 1.

The results for the time $t = 100$ and three different values of disorder are shown in Fig. 5. On the ergodic side ($W = 4$), the distribution is broader than for stronger disorder, with lower peaks and a higher "valley." In order to visualize the dynamics of the spin distributions, we show them as a function of time in Fig. 6 in the form of a contour plot. This representation demonstrates clearly qualitative differences between the ergodic and localized regimes. For $W = 4$ one can see the peaks close to $\langle S^z \rangle = \pm 1$ slowly fading, whereas they are persistent in the cases $W = 5$ and $W = 8$. The persistence of peaks is a signature of localization analogous to the saturation of the imbalance (3). In the strongly localized case, $W = 8$, barely any dynamics in $\mathcal{P}(\langle S^z \rangle)$ is visible.

## IV. CONCLUSIONS

We have investigated many-body localization in the quench dynamics of a quasi-periodic spin chain at late times for moderately large chains up to $L = 50$, using the time-dependent variational principle applied to matrix product states. Using an initially antiferromagnetically ordered state, we have determined the time evolution of spin densities for various choices of the quasi-periodic field strength (disorder strength) $W$ and periodicity parameter $\Phi$. Since the TDVP method used here is the same as used in Ref. [25] for the case of purely random disorder, we can efficiently compare the quasi-periodic and random cases. Our key findings can be summarized as follows.

First, our results demonstrate that the critical strength $W_c$ of the quasi-periodic field corresponding to the MBL

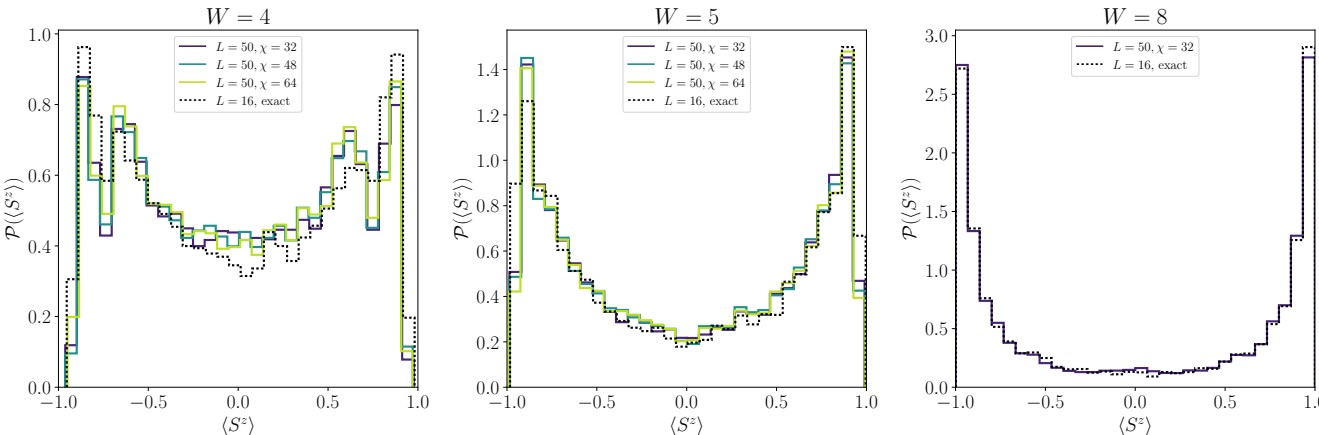

FIG. 5. Normalized distribution of $\langle S^z \rangle$ over all lattice sites and realizations of disorder at a fixed time $t = 100$, for various choices of the disorder strength $W$. The histogram uses 30 equidistant bins.

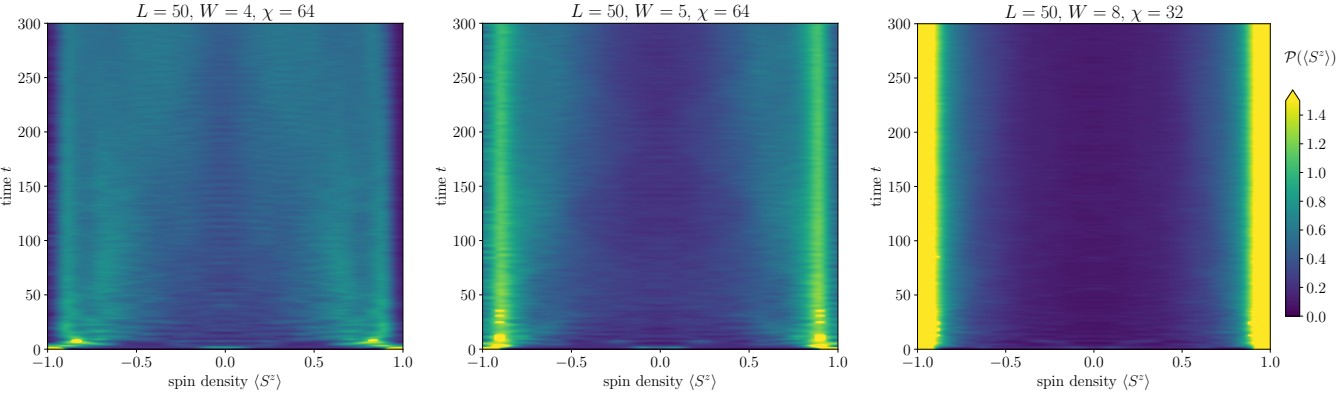

FIG. 6. Interpolated filled contour plot of the distribution of $\langle S^z \rangle$ over all lattice sites and realizations of disorder as a function of time, for various choices of the disorder strength $W$. The color indicates the value of $\mathcal{P}(\langle S^z \rangle)$. Each "slice" in time corresponds to a histogram as in Fig. 5 with 30 equidistant bins. Values of $\mathcal{P}(\langle S^z \rangle) > 1.5$ are indicated with the same color.

transition at given interaction strength ($\Delta = 1$) depends on $\Phi$ in an essential way, as opposed to the non-interacting case (the Aubry-André model), where the transition is at $W = 2$ for almost any irrational $\Phi$. Specifically, we find $W_c \simeq 4.8 \pm 0.5$ for $\Phi = (\sqrt{5}-1)/2$ and $W_c \simeq 7.8 \pm 0.5$ for $\Phi = \sqrt{2}/2$. The dependence of $W_c$ on $\Phi$ is qualitatively explained by considering the statistical properties of neighboring values of the potential [47]. This influences properties of the corresponding localized single-particle states, and, as a result, the susceptibility of the system to the many-body-delocalizing effect of the interaction.

It is worth mentioning that the analysis of the imbalance decay is complicated by the appearance of slow oscillations, which are found to be related to the long-range correlations that occur in quasi-periodic potentials. This effect might be alleviated by averaging over randomized initial conditions instead of the fixed anti-ferromagnetic initial state. On the other hand, randomizing the initial state can itself be related to rare events [12], which might obscure the physics of the quasi-periodic model.

Our second main conclusion concerns the value of the critical disorder $W_c$ and the finite-size effects. We find, by comparing the TDVP data for $L = 50$ to exact numerical results for a relatively small system with $L = 16$, that finite-size effects on the critical disorder strength are considerably weaker than in the purely random case. Indeed, contrary to the case of purely random disorder, our result for the critical disorder $W_c \simeq 4.8 \pm 0.5$ for $\Phi = (\sqrt{5}-1)/2$ and $W_c \simeq 7.8 \pm 0.5$ for $\Phi = \sqrt{2}/2$ do not depend significantly on system size (see Fig. 3). This is consistent with conclusions of Ref. [23] and, to our knowledge, provides the first direct evidence of a quantitative difference in the severity of finite-size effects for the two disorder scenarios beyond the small systems studied with exact diagonalization. A possible explanation for this difference between the random and quasi-periodic models is the appearance of extended, rare thermal regions in the purely random case, which can act as baths for global delocalization of the system in accordance with the "quantum avalanche" scenario [20, 21]. In a quasi-periodic system, this mechanism of delocalization by rare regions is not operative. It

also worth emphasizing that the found values of $W_c$ are substantially lower than $W_c \approx 11$ (in units of this paper) obtained for fully random potential [25]. This can be explained again by the effect of rare "ergodic spots" enhancing the delocalization in the case of purely random disorder.

On the other hand, we find that increasing system size leads to a modest increase in the rate of delocalization on the ergodic side, as measured by the decay of the imbalance, in agreement with Ref. [12]. This behavior is similar to the one found in the case of purely random disorder [25]. Hence, while there do not appear to be strong finite-size effects on the strength of the quasi-periodic field $W_c$ that separates the ergodic and non-ergodic regimes, there are still noticeable finite-size effects on how quickly the system thermalizes.

Finally, we have investigated the robustness of power laws in the imbalance decay, as found in experimental results for relatively short times [11, 12] and in the exact-diagonalization numerics [26]. For this purpose, we have considered the flow of the "running power-law exponent" $\beta(t)$ with time $t$. For short times, as well as for all times in small systems ($L = 16$), we do find a power-law behavior, in consistency with Refs. [11, 12, 26]. On the other hand, the imbalance decay in a longer system, $L = 50$, exhibits an increase in the decay rate with time, at variance with the power-law behavior in the purely random case. The requirement of convergence limits the range of accessible times, in which the effect remains relatively modest. Our data suggest, however, that the increase of the decay rate will likely accelerate for still longer times, in agreement with Hartree-Fock simulations of Ref. [27].

It remains a challenge for future MPS-based calculations (or related approaches) to push the controllable analysis of the dynamics to still longer times in order to find the long-time dependence of the imbalance and other observables on the ergodic side of the transition. It will be also interesting to see whether shift-inverse exact diagonalization technique [53], which can access systems up to $L = 26$, will be sufficient to detect deviations from the power-law behavior. Another prospective direction would be to extend our analysis to the related interacting Fibonacci quasi-periodic model [54]. In that paper, it was argued that, remarkably, adding interacting to this system does not enhance delocalization. It would be interesting to investigate the influence of larger system sizes on this phenomenon.

Experimentally, the system sizes probed in this work can be assessed using cold atoms [12] or ion traps [55]. The system length $L = 50$ that we used is a typical number for these experiments. Thus, an experimental setup that sufficiently suppresses noise and particle loss over times $t > 100$ ought to be able to confirm our findings. The analysis that we have performed, both for the imbalance and the spin density distribution, can be straightforwardly applied to experimental data in order to analyze and visualize the dynamics of many-body delocalization.

Two related preprints appeared after the preprint version of this manuscript was posted on arXiv. Ref. [56] studies similarities and differences between the quasi-periodic case and the purely random case by using exact diagonalization and inspecting the behavior of a running exponent $\beta(t)$ (distinct from our $\beta(t)$) defined from the spatial spreading of the diffusive propagator with time, similarly to a previous work studying the purely random case [57]. The focus of Ref. [56] is on the law of approach of $\beta(t)$ to zero on the MBL side of the transition, i.e., is different from that of our paper. Ref. [58] studies the interacting Aubry-André model using a combination of t-DMRG ($\chi = 40, t \leq 120$), exact diagonalization, and machine learning, and finds an estimate $W_c \approx 3.8$ for the MBL transition at $\Delta = 1/2$. This compares well with our result $W_c \approx 4.8$ for $\Delta = 1$ and the result $W_c \approx 3$ as obtained for $\Delta = 1/4$ [26].

## ACKNOWLEDGEMENTS

We thank I. Bloch for useful discussions and suggesting to study this model, M. Knap for useful discussions and suggesting to consider the distribution plotted in Fig. 2, and M. Žnidarič for useful discussions. TDVP simulations were performed using the open-source evoMPS library [59], implementing the single-site algorithm of Ref. [39]. We used Matplotlib [60] to generate figures. The authors acknowledge support by the state of Baden-Württemberg through bwHPC.

## Appendix A: Numerical details

In our simulations, we employ the time-dependent variational principle (TDVP) as applied to matrix product states. The TDVP was first developed by Dirac [61] as a means of approximately solving the time-dependent Schrödinger equation, given some variational ansatz. Realizing that a matrix product state boils down to just a variational ansatz within the exponentially large many-body Hilbert space, Haegeman *et al.* applied this idea to MPSes [38]. However, the algorithm was not immediately widely adopted since in most cases it does not provide clear benefits over established time-dependent methods for one-dimensional systems, such as time-evolving block decimation [31]. A significant advance was made when, in a follow-up paper, Haegeman *et al.* published a refined version of their algorithm, *split-step single-site TDVP* (in the language of the recent review of MPS methods [43], 1TDVP). This algorithm has the considerable benefit that the projection onto the MPS manifold conserves quantities that are conserved by the Hamiltonian dynamics, such as the global energy in our Hamiltonian (1).

The development of 1TDVP suggested its use as an *approximate* method for time evolution. Leviatan *et al.* [62] have argued that 1TDVP can yield accurate results for averaged transport properties (e.g. diffusion constants) even with a very small bond dimension, the use of which

would yield obviously wrong results for TEBD (and other older MPS methods) due to the non-conservation of energy alone. However, Kloss *et al.* [41] cautioned against trusting the results of TDVP in such a scenario, which they showed yields incorrect results for some systems in terms of transport-related quantities. On the other hand, they also showed that for certain systems—in particular, the disordered XXZ chain—the TDVP yields reasonable results even at times much longer than those where "traditional" MPS methods would fail to converge. These findings were confirmed in a later work by us with co-authors [25].

With the reliability of 1TDVP at intermediate timescales confirmed for the disordered XXZ chain (at least for not too weak disorder), it is natural to expect it is also applicable to the quasi-periodic XXZ chain (1) close to the MBL transition. Here we show several additional benchmarks to confirm that this is indeed the case. We refer the interested reader to the appendix of the previous work [25] for further performance checks pertaining to the case of purely random disorder.

One peculiarity of the 1TDVP algorithm is that the exact projection onto the MPS manifold only applies to time evolution with a fixed bond dimension. In our implementation, we start with a product state with $\chi = 1$, and then expand the variational manifold to the desired value using the Euler-forward time integrator, which is complete during the early time evolution. The Euler-forward integrator does not conserve energy (see Fig. 7d), so in practice we take a tiny step using Euler-forward to expand the size of the variational manifold, followed by a step of size $\delta t$ with (the energy-conserving) split-step 1TDVP. This procedure is repeated until the desired bond dimension $\chi$ is reached. The remaining time evolution is then computed using only split-step 1TDVP. Hence, 1TDVP does not have the adaptive nature of the more common two-site approaches, losing some numerical efficiency, and convergence can only be established by considering results at different bond dimension.

The numerical cost of 1TDVP is also less predictable than in the case of e.g. TEBD, which uses an explicit time integrator as opposed to the requirement of solving a set of equations for the implicit 1TDVP algorithm. We find that for our (not necessarily optimal) implementations, the computational cost for 1TDVP using $\chi = 64$ is comparable to the computational cost of TEBD using $\chi = 128$. It is important to stress, however, that both methods should not be compared only based on the bond dimension. Indeed, we find that for the system under study, TDVP with a low bond dimension is much more reliable than TEBD with a higher bond dimension, at least when it comes to transport properties.

As a first check, we compare dynamics of 1TDVP with $\chi = 64$ to the TEBD algorithm, where we use the OSMPS library [63, 64] for our TEBD implementation, using default convergence parameters and the same time-step $\delta t = 0.1$. The results are shown in Figure 7. As expected, the results match closely at short times, where

the accumulated truncation error for TEBD and the projection error for TDVP are negligible and both methods can be regarded as numerically exact. The TEBD algorithm reaches its maximum set bond dimension already at $t \approx 15$ for $\chi = 128$; for $\chi = 256$ it fares only slightly better, reaching the maximum bond dimension at $t \approx 25$. Significant deviations with TDVP results become apparent at $t \approx 100$, which coincides with the time at which substantial errors in the energy arise. Specifically, the TEBD result for the imbalance saturates, while the TDVP results continue to decay. The TEBD result for $\chi = 256$ follow the TDVP results until slightly longer time but then again saturates. This is again in correspondence with the fact the error in energy in the TEBD result for $\chi = 256$ accumulates slightly more slowly than for $\chi = 128$. As expected, 1TDVP yields no significant errors in the energy.

As an additional illustration of the difference between TDVP and TEBD, we compare in Fig. 8 the behavior of the time-dependent "flowing power law" $\beta(t)$, computed in the same way as in Sec. III B. The TDVP results with $\chi = 128$ have nearly converged up to the time $t = 200$ (cf. Fig. 4 where we showed full convergence when only data up to $t = 180$ are considered). On the other hand, the TEBD data with $\chi = 256$ in this range of $t$ are still quite far from convergence despite a larger value of the bond dimension. One clearly sees that the TEBD data start to develop a slight increase of $\beta(t)$ at this value of $\chi$, approaching the TDVP results. At longer time, the TEBD results show a downturn in $\beta(t)$ towards 0 reflecting the spurious saturation of the imbalance. With increasing $\chi$, the time at which this downturn starts becomes longer.

We thus see that the TDVP exhibits superior performance in comparison to TEBD from the point of view of convergence (and thus reliability) of transport observables. Below we discuss some further aspects of both methods. This will allow us to argue that the better performance of TDVP is related to energy conservation in this approach.

As one of possible measures of convergence, one often considers the so-called discarded weight, an estimate of the error induced by the cutoff $\chi$ in the entanglement spectrum. The latter quantity can be obtained from the "entanglement Hamiltonian" [65, 66], in turn related to the reduced density matrix $\rho$ by tracing out one part of the system in a bipartition. For such a bipartition into subsystems $A$ and $B$, the entanglement Hamiltonian $H_e = -\ln \text{Tr}_B \rho$. The squared values of the eigenvalues of $H_e$ make up the entanglement spectrum: $\{\lambda_1, \ldots, \lambda_\chi\}$ (ordered in a descending way, and truncated in the numerical procedure after $\lambda_\chi$). In TEBD, the discarded weight can be estimated using the singular value decomposition employed during the algorithm, which yields an approximation of the discarded values $\sum_{i=\chi+1}^{\mathcal{N}} \lambda_i$, where $\mathcal{N}$ is half the size of the Hilbert space for a bipartition in the middle of the chain [40]. However, there is no discarded weight in the single-site approach of 1TDVP, which does not employ an explicit truncation of

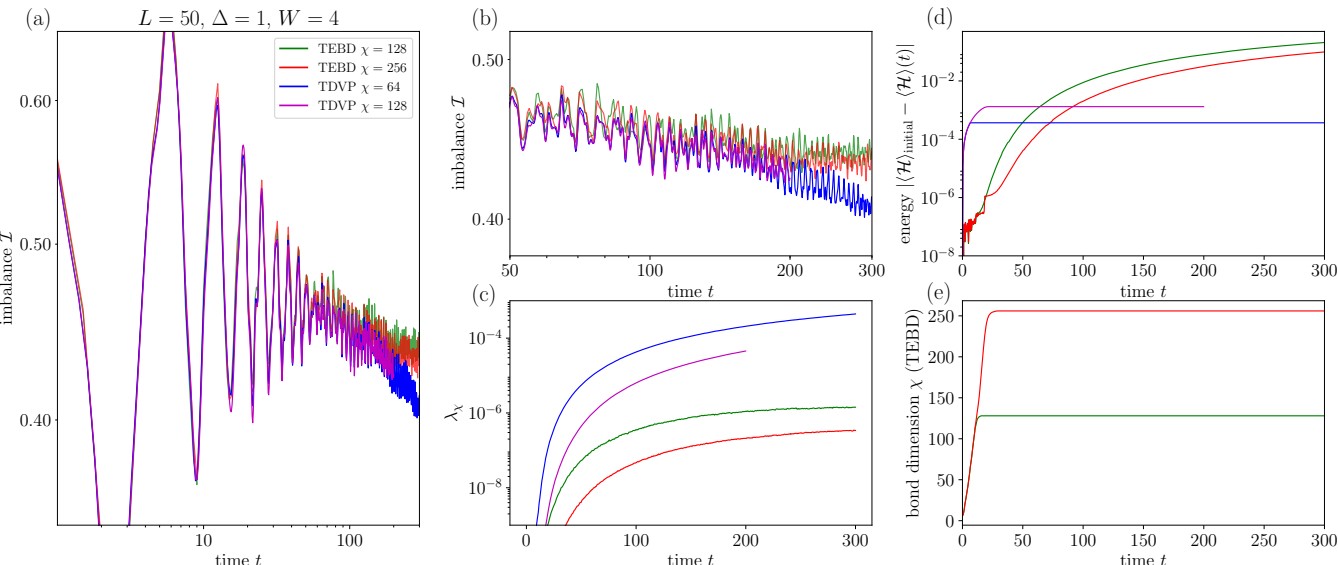

FIG. 7. Comparison of TDVP and TEBD results for $W = 4$, $\Phi = (\sqrt{5} - 1)/2$, $L = 50$. **(a)-(b)**: disorder-averaged imbalance for $\chi = 64$ (TDVP), $\chi = 128$ (TDVP, TEBD) and $\chi = 256$ (TEBD). TEBD results are averaged over approximately 150 realizations, TDVP results are averaged over 400 realizations for $\chi = 64$ and $\chi = 128$ respectively. **(c)**: disorder-averaged smallest value in the entanglement spectrum. **(d)**: disorder-averaged energy $\langle \mathcal{H} \rangle$, where the difference between the energy and the initial value is shown. **(e)**: bond dimension as a function of time for the TEBD implementation.

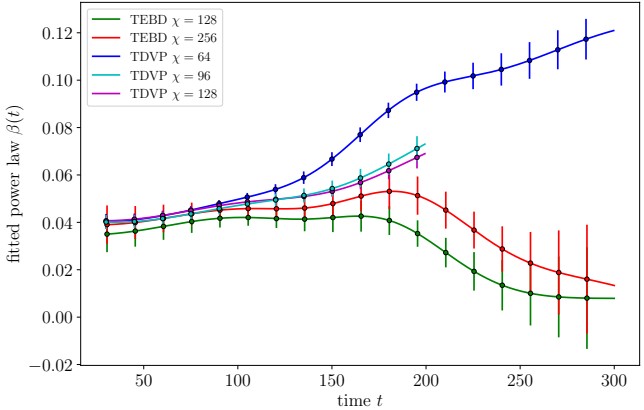

FIG. 8. Comparison of the behavior of $\beta(t)$, cf. Sec. III B, for both TDVP and TEBD for the same parameters as in Fig. 7: $W = 4$, $\Phi = (\sqrt{5} - 1)/2$, $L = 50$. Colour coding is as in Fig. 7; we have also included TDVP data for $\chi = 96$. Error bars are $2\sigma$-intervals based on a statistical bootstrap.

the entanglement spectrum, unlike TEBD. To compare the methods, we consider a related quantity: the smallest value $\lambda_\chi$. The results are shown in Fig. 7c. We see that, from the point of view of this characteristics, the performance of TEBD is better than that of 1TDVP with the same bond dimension. This indicates that 1TDVP is not as favorable for determining long-time entanglement properties as it is for transport properties, which has been previously observed in Ref. [25]. It should be emphasized, however, that this seemingly better perfor-

mance of TEBD is somewhat deceptive: the small value of $\lambda_\chi$ does not imply a correct behavior since the errors accumulate (the accumulated $\lambda_\chi$'s should behave in a qualitatively similar manner to the accumulated discarded weight of TEBD and t-DMRG). As an example, we see from Fig. 7c,d that, although $\lambda_\chi$ remains $\lesssim 10^{-7}$ up to times $t \sim 300$ for TEBD with $\chi = 256$, the corresponding error in energy turns out to be as large as $10^{-1}$ and the imbalance shows a spurious saturation. Meanwhile, the error in the energy for TDVP is only due to the early time steps using the Euler-forward method, remains at a bounded, modest value $< 10^{-3}$, and the energy is correctly conserved at late times.

As a further demonstration of reliability of the TDVP, we consider its results for a single realization with different bond dimension. In Figure 9 (left panel) we show time dynamics for a single choice of the random phase $\phi_0 = 0$ for different $\chi = 32$ and $\chi = 64$. One observes that at short times, the two curves are indistinguishable on the scale of the figure until $t \approx 120$. In the regime $t \gtrsim 120$, small differences become apparent, which increase over time. The time until which the full convergence persists is essentially the same as obtained for the "flowing power law" of the averaged imbalance for the same value of bond dimension, see the left panel of Fig. 4. As was pointed out above, this time increases up to $t \approx 180$ when the bond dimension is raised up to $\chi = 128$. Thus, the convergence of imbalance as given by the TDVP within these time intervals holds not only for the average but also for a single realization of disorder.

It is also important to check the convergence with the time-step of the integrator, which is shown in Fig. 9 (right

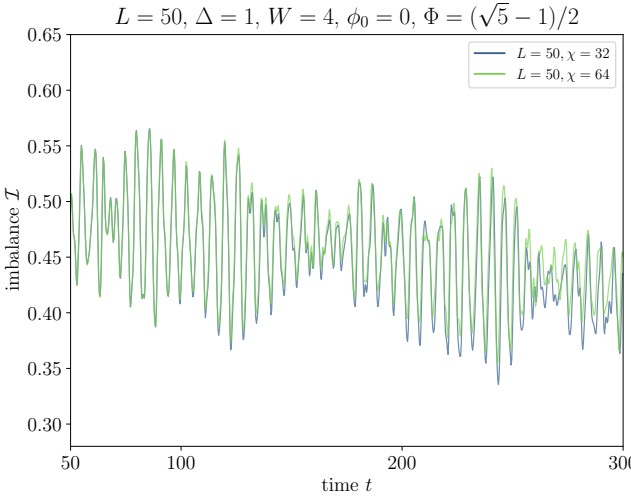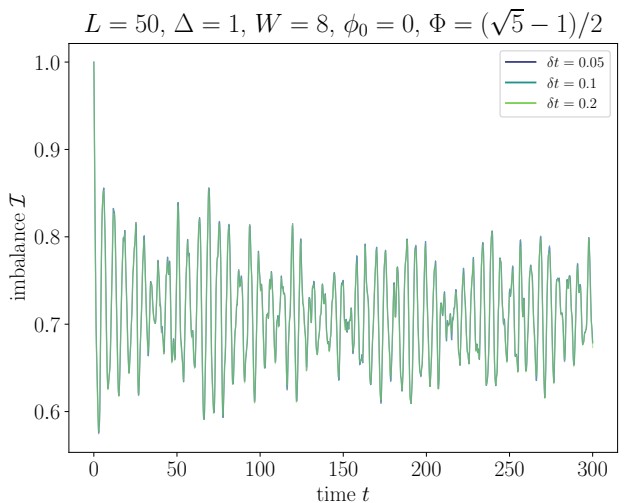

FIG. 9. Left: time evolution of the imbalance (3) as obtained by the TDVP for a single disorder realization: the specific choice of phase $\phi_0 = 0$, at times $t > 50$, using different choices of bond dimension $\chi = 32$ and $\chi = 64$. The other parameters are: the strength of disorder $W = 4$, periodicity $\Phi = (\sqrt{5} - 1)/2$, and system size $L = 50$. Right: convergence of TDVP with the size of the time step $\delta t$ for $W = 8$, $\chi = 32$, and otherwise the same parameters as in the left panel. The lines with the time steps $\delta t = 0.05$, $0.1$, and $0.2$ are essentially indistinguishable.

panel) for the case $W = 8$. As is typical of implicit time integration methods, TDVP works very well even for a sizeable time step, and the results for $\delta t = \{0.05, 0.1, 0.2\}$ exhibit no significant differences.

Summarizing, the superior performance of TDVP when compared to TEBD can be understood as a consequence of energy conservation. In the regime where the methods are no longer numerically exact, TEBD yields non-conservation of energy and the result quickly becomes unphysical. For example, studying the $W = 4$

system with TEBD, one would conclude that the imbalance saturates and thus the system is localized, while in reality it is well on the delocalized side of the transition. On the other hand, TDVP is forced to have the correct conserved quantities, which explains why it works exceptionally well in describing transport properties for disordered systems, compared to other MPS methods. Of course, TDVP is also not exact. Eventually, the added noise due to the semiclassical approximation inherent in 1TDVP accumulates, and convergence gets lost at some time which is a monotonously increasing function of $\chi$.

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
