# Peer review of "Many-body delocalization dynamics in long Aubry-André quasi-periodic chains"

_SciPost Physics_

## Round 1 · Referee Report · Anonymous · 2019-2-15

Strengths

1. Application of time-dependent variational technique with matrix product states to the problem of MBL with quasi-periodic disorder
2. Introducing distribution of spin densities as a possible experimental observable.

Weaknesses

1. Results seem not to be converged - that makes the claims of faster than power-law decay of imbalance doubtful.
2. The effect found and discussed, the dependence of the system behavior on periodicity of quasi-periodic potential is not new.

Report

The manuscript addresses many-body localization (MBL) in quasi-periodic on-site disorder in isotropic Heisenberg chain model - a paradigmatic model used for MBL studies. While the manuscript lists some of earlier works on this very topic [20,23,26] it misses, however, at least one early study PRB 87, 134202 (2013). Those earlier studies used typically exact diagonalization combined with the finite size scaling to estimate the transition to MBL. The present work considers experimentally accessible observables such as imbalance and distribution of spin densities whose time dynamics is evaluated using time-dependent variational principle with matrix product states. The authors report as their main finding (abstract and conclusion) the strong dependence of the system properties on the periodicity of the quasi-periodic potential which they trace to properties of the non-interacting problem. They stress that the decay of the anti-ferromagnetic order is faster than a power law (I believe on the delocalized side of the MBL transition).

The field of MBL is quite well established by now so making a significant contribution in the field is not as straightforward as say 5 years ago. In my opinion the work presented while providing some incremental value to the field, certainly does not meet the Scipost Physics criterion: "Articles provide in-depth, detailed reports of groundbreaking research within one or more subject areas."

The main claim, a strong dependence of the system dynamics on the periodicity of the quasi-periodic potential is a bit trivial and not new. It is explained more than 10 years ago in detail in V. Guearrera et al. New. J. Phys. 9, 107 (2007). In fact the on-site energy difference distribution (plotted in Fig.2 right panel of the manuscript) may be easily found analytically as proportional to $1/\sqrt{1-x^2}$ distribution with the amplitude of the effective disorder $W\sin(\pi\Phi)$ as it follows from that paper. Similar strong dependence on the quasi-periodicity was found in an interaction free problem by Major et al. Phys. Rev. A 98 053633 (2018).

While the application of the variational principle MPS approach to time dynamics seems to be an original contribution, this has been already reported in [22] for a purely random case. So one can hardly give credit for that. Especially, as time dependent dynamics using MPS (with e.g. TEBD) for MBL has been studied in a routine way, see [8], also
J. H. Bardarson, F. Pollmann, and J. E. Moore, Phys. Rev. Lett. 109, 017202 (2012)
P. Sierant, D. Delande, and J. Zakrzewski,Phys. Rev. A 95, 021601(R) (2017)
for different implementations.

The variational approach might have some advantage for long times due to the inherent energy conservation, the problem deserves a separate careful study in the MBL regime and close to it - comparisons as those in 1901.05824 do not suggest a clear advantage of the variational approach.

There is a strong believe that the system studied shows the mobility edge - the transition to MBL depends on energy (see Luitz et al Phys. Rev. B 91, 081103 (2015)). The present study considers a single initial anti-ferromagnetic state whose energy depends strongly on the disorder realization. This may affect the value of the critical disorder obtained. The question of the mobility edge is entirely ignored in the manuscript.

Apart from these general remarks I believe that the authors should (in some future submission) consider the following remarks
1. Results for $\chi=32$ and $\chi=64$ seem unconverged (Fig. 1a and Fig.1b). Why not show convergent results for sufficiently large $\chi$?
2. The present Fig.1 is unreadable - it requires huge enlarging to distinguish different curves.
3. Let me mention that standard TEBD results with $\chi=250$ do not agree with those reported in Fig.1.
4. Why the authors do not provide information that would allow a reader to reproduce their results? What was the time step in time-dependent integration? What was the accuracy criterion (none - just $\chi$ was defined is one of the options)?
5. The faster than power-law decay of the imbalance on the delocalized side, the main conclusion of the manuscript, is deduced from not converged results.
6. The claim for the critical disorder to be $W_c=5$ is a rough estimate only. The text suggest error bar of the order of 0.5. Can the authors do better with the converged results?
7. Authors claim a weak dependence on the system size for quasi-periodic potential in the introduction but they do not substantiate this claim further - showing just a single curve for $L=16$. How $W_c$ depends on the system size?

Requested changes

see above

  • validity: low
  • significance: low
  • originality: ok
  • clarity: ok
  • formatting: good
  • grammar: perfect

Author Elmer Doggen on 2019-03-07 (in reply to Report 1 on 2019-02-15)
Category:
answer to question
reply to objection

Following a possibility provided by SciPost, we provide a preliminary response to the Referee's comments (Report 1). We will finalize the response and resubmit the paper, once the Editor-in-Charge invites us to do so upon the end of the refereeing round.

We thank the Referee for a detailed report. The referee's comments motivated us to improve the rigour of our results with some additional numerical computations, which will be added to a future submission. In addition, we will improve the presentation of our work and the clarity of our plots, as well as providing more details about our numerical method and the benefits thereof compared to older methods like time-evolving block decimation (TEBD). On the other hand, we disagree with the referee's assessment of our work as having low significance and validity. We address all critical remarks of the referee below.

REFEREE:
"The manuscript addresses many-body localization (MBL) in quasi-periodic on-site disorder in isotropic Heisenberg chain model - a paradigmatic model used for MBL studies. While the manuscript lists some of earlier works on this very topic [20,23,26] it misses, however, at least one early study PRB 87, 134202 (2013)."

We thank the referee for pointing out this reference that is indeed relevant - a future submission will include a citation.

REFEREE:
"Those earlier studies used typically exact diagonalization combined with the finite size scaling to estimate the transition to MBL. The present work considers experimentally accessible observables such as imbalance and distribution of spin densities whose time dynamics is evaluated using time-dependent variational principle with matrix product states. The authors report as their main finding (abstract and conclusion) the strong dependence of the system properties on the periodicity of the quasi-periodic potential which they trace to properties of the non-interacting problem. They stress that the decay of the anti-ferromagnetic order is faster than a power law (I believe on the delocalized side of the MBL transition)."

There are in fact three main conclusions that are listed clearly in the summary and are mentioned also in the abstract: the dependence of the MBL transition on $\Phi$, the faster-than-power law decay on the delocalized side and the relatively weak dependence of the transition on system size. We will improve the wording in the abstract to make this clearer.

REFEREE:
"The field of MBL is quite well established by now so making a significant contribution in the field is not as straightforward as say 5 years ago. In my opinion the work presented while providing some incremental value to the field, certainly does not meet the Scipost Physics criterion: "Articles provide in-depth, detailed reports of groundbreaking research within one or more subject areas." "

We disagree with this assessment. In our view our study provides an important addition to the existing literature by studying larger systems up to late times, which is of obvious relevance to current experiments. The differences between the MBL transition in quasiperiodic and purely random systems is currently under active discussion and our work provides much needed numerical analysis to help settle this debate.

REFEREE:
"The main claim, a strong dependence of the system dynamics on the periodicity of the quasi-periodic potential is a bit trivial and not new. It is explained more than 10 years ago in detail in V. Guearrera et al. New. J. Phys. 9, 107 (2007). In fact the on-site energy difference distribution (plotted in Fig.2 right panel of the manuscript) may be easily found analytically as proportional to $1/\sqrt{1−x^2}$ distribution with the amplitude of the effective disorder $W\sin(\pi \Phi)$ as it follows from that paper. Similar strong dependence on the quasi-periodicity was found in an interaction free problem by Major et al. Phys. Rev. A 98 053633 (2018). "

We thank the referee for pointing out the important reference New. J. Phys. 9, 107 (2007), of which we were not aware. The effective disorder mentioned in that work matches with our Fig. 2b after correcting with an additional factor of 2 due to their potential using a squared sine. Our "main claim" is not about the single-particle problem; indeed, that the periodicity strongly affects dynamics was pointed out, as mentioned in our manuscript, also in Ref. [36], Phys. Rev. A 80, 021603 (2009) for the extended AA model. Rather, one of the key results is that this single-particle feature carries over to the many-problem directly and leads to an apparent dependence of the critical disorder on the periodicity $\Phi$ - a non-trivial and to our knowledge new result contrary to the non-interacting case where the critical disorder is always $W = 2$ (in our units) for almost any irrational $\Phi$. That there are still interesting results from recent publications such as in [Phys. Rev. A 98 053633] even for the non-interacting problem underscores that there is still much to learn in the interacting problem.

REFEREE:
"While the application of the variational principle MPS approach to time dynamics seems to be an original contribution, this has been already reported in [22] for a purely random case. So one can hardly give credit for that. "

In Ref. [22] we demonstrated that our method works well for the MBL problem with purely random disorder, allowing to reliably study large systems and long times on the ergodic side of the transition. Hence, it is natural and important to use it also for the quasi-periodic case and compare the results.

REFEREE:
"Especially, as time dependent dynamics using MPS (with e.g. TEBD) for MBL has been studied in a routine way, see [8], also
J. H. Bardarson, F. Pollmann, and J. E. Moore, Phys. Rev. Lett. 109, 017202 (2012)
P. Sierant, D. Delande, and J. Zakrzewski,Phys. Rev. A 95, 021601(R) (2017)
for different implementations.

The variational approach might have some advantage for long times due to the inherent energy conservation, the problem deserves a separate careful study in the MBL regime and close to it - comparisons as those in 1901.05824 do not suggest a clear advantage of the variational approach."

We thank the referee for pointing out these references, which will be cited appropriately. Our method provides considerable advantages over TEBD specifically for the MBL problem (see also our response to point 3 below). Indeed, the main advantage of the method is the inherent energy conservation, as opposed to the non-conservation of energy in e.g. TEBD. As a side note, in the language of 1901.05824 we use a 1TDVP algorithm, not a 2TDVP algorithm. The latter does not conserve energy for otherwise energy-conserving Hamiltonian dynamics due to the truncation step induced by the two-site approach.

REFEREE:
"There is a strong believe that the system studied shows the mobility edge - the transition to MBL depends on energy (see Luitz et al Phys. Rev. B 91, 081103 (2015)). The present study considers a single initial anti-ferromagnetic state whose energy depends strongly on the disorder realization. This may affect the value of the critical disorder obtained. The question of the mobility edge is entirely ignored in the manuscript."

A dependence of the critical disorder on energy in this system is indeed interesting, but we do not expect the essential properties of the MBL transition to depend on energy (as long as we are not too close to the ground state). A large majority of studies of MBL only target the middle of the spectrum and likewise ignore the question of a mobility edge. We thus leave the investigation of the energy-dependence of the critical point and verification of the universal character of the transition to future work.

REFEREE:
"Apart from these general remarks I believe that the authors should (in some future submission) consider the following remarks
1. Results for $\chi=32$ and $\chi=64$ seem unconverged (Fig. 1a and Fig.1b). Why not show convergent results for sufficiently large $\chi$?"

We might have shown only $\chi = 64$ and terminate the lines where convergence with bond dimension is lost, but opted to show results for both $\chi = 32$ and $\chi = 64$ for transparency. We believe that this additional information is useful for a reader. Concerning the value of $\chi$: we find that our TDVP implementation is slower than TEBD using the same $\chi$, so we cannot compute results for these long times with $\chi$ much higher than $64$ with realistic computational resources. This is due to the implicit time integration scheme used in the split-step 1TDVP algorithm, meaning we have to solve a set of equations at each time step. For this purpose evoMPS uses an iterative scheme, which does not always converge quickly. This is a "price" that one pays for the energy conservation in this scheme. The energy conservation, is, however, very important: the energy-non-conserving TEBD scheme becomes unreliable at shorter times (despite large values of $\chi$) - at least for this system - see our response to question 3 below.

REFEREE:
"2. The present Fig.1 is unreadable - it requires huge enlarging to distinguish different curves. "

Indeed, in each of the panels of Fig. 1 there are two curves that lay essentially on top of each other. In the left ($W=4$) and middle ($W=5$) panel this happens because the convergence with bond dimension has been reached, and we want to demonstrate this. In the right ($W=8$) panel these are two curves with lengths $L=16$ and $L=50$; here this demonstrates very weak length dependence. We will update our figures in a resubmission to enhance clarity (by adding some insets) and will also make clarifying comments in the figure caption.

REFEREE:
"3. Let me mention that standard TEBD results with $\chi=250$ do not agree with those reported in Fig.1."

Indeed, the results from TEBD and TDVP on the delocalized side of the transition begin to deviate for long times. This is related to the non-conservation of energy in the TEBD method.
Motivated by the referee's comment, we have computed dynamics for $\Phi = [\sqrt{5}-1]/2$ and $W=4$ (delocalized side of the transition) using TEBD with $\chi = 128$ and $\chi = 256$. The plots are added as attachments, with TDVP results ($\chi = 64$) shown in blue and TEBD results in green and red. In addition to the imbalance, we show there also the energy as well the bond dimension used by TEBD (which saturates at the maximum value, which is 128 for one plot and 256 for the other).
Here the TEBD results are computed using the OSMPS library using default convergence parameters using around 150 realizations. As is visible from the plots, TEBD results start to deviate significantly from TDVP exactly where large errors in the energy become apparent in the TEBD results. The corresponding time is $t \approx 70$ for TEBD bond dimension 128, and somewhat later for TEBD bond dimension 256, so that the TEBD results are reliable for longer times when TEBD bond dimension is increased, as expected.

REFEREE:
"4. Why the authors do not provide information that would allow a reader to reproduce their results? What was the time step in time-dependent integration? What was the accuracy criterion (none - just $\chi$ was defined is one of the options)?"

As was briefly mentioned in the manuscript, our implementation of the method follows Ref. [22] and the bond dimension is indeed the only parameter used to establish convergence. We agree, however, with the Referee that a more detailed exposition of the method will be beneficial for the reader. We will provide some more details and references about the numerical method and how it compares to "traditional" MPS methods in a resubmission. The time-step used is 0.1 in lattice units. We use open source MPS libraries and are happy to share our implementations thereof upon reasonable request.

REFEREE:
"5. The faster than power-law decay of the imbalance on the delocalized side, the main conclusion of the manuscript, is deduced from not converged results."

The faster-than-power law decay is apparent in the converged results. In Fig. 3 we show the "flowing power law" $\beta$, which is an effective time-dependent power law. As is seen from the results, for $\Phi = (\sqrt{5}-1)/2$ convergence with $\chi$ is achieved up to $t \approx 200$, and a clear increase in $\beta(t)$ (roughly by a factor 2.5) is observed already by this time. Note that we did not observe this trend for purely random disorder in Ref. [22], so that we see a clear difference between purely random and quasiperiodic systems within the same approach. The non-converged results in the left panel of Fig. 1 also show a strong curvature, i.e. a faster-than-power law decay. While it is expected that the curvature remains when the converged is reached, we do not draw any conclusions from those results. We will further clarify this in the text in the new version.

REFEREE:
"6. The claim for the critical disorder to be $W_c=5$ is a rough estimate only. The text suggest error bar of the order of 0.5. Can the authors do better with the converged results?"

We still find a very weak decay of imbalance for $W = 4.5$, $L = 50$, and no visible decay for $W=5$, which yields the estimate for the critical disorder and the error bar of order 0.5. It is very difficult to improve the error bar substantially and, at the same time, reliably. In fact, 10% is a pretty good accuracy for determination of the MBL transition. We remind the referee that, for the purely random case, the best value obtained by Luitz et al. from exact diagonalization was $\approx 3.7$, whereas the true transition (based on our previous results in Ref. [22]) is in fact near 5.5, i.e. at 50% higher disorder. In many cases, small error bars quoted in the literature do not correspond to reality.

REFEREE:
"7. Authors claim a weak dependence on the system size for quasi-periodic potential in the introduction but they do not substantiate this claim further - showing just a single curve for $L=16$. How $W_c$ depends on the system size?"

In the case of purely random disorder, the apparent position of the transition changed from $W = 3.5-4$ for $L = 16$ to $W \approx 5.5$ using $L=50$ and $L = 100$, which is a 50% increase. For the quasi-periodic case, we do not observe such a strong increase: the value remains the same within 10% accuracy. We will add a plot to a future resubmission showing more results for different $L$ and $W$, to show this more clearly.

Attachment:

tebd_comparison.pdf

---

## Round 1 · Referee Report · Anonymous · 2019-3-21

Strengths

1- Application of TDVP to the quasiperiodic (QP) interacting system

Weaknesses

1- Proper convergence of the method with time-step and bond-dimension is not established
2- Conclusion about the drift of the dynamical exponent is drawn from a regime were the data doesn't appear to be converged
3- The study doesn't bring much new information compared to literature.
4- Poor presentation, which doesn't allow to read the results conveniently

Report

The authors numerically study the dynamics in a quasiperiodic (QP) interacting spin chain, which shows a MBL like transition as a function of the strength of the QP potential.

In their introduction the authors claim:

"The aim of the present work is to investigate the differences between purely random
and quasi-periodic disorder at system sizes inaccessible to exact diagonalization, using the
newly developed numerical technique of the time-dependent variational principle as applied
to matrix product states"

This statement is however misleading, since dynamics in QP interacting systems was studied using MPS based method in both [37] (L=100-200) and 10.1073/pnas.1800589115, for systems of up to 800 sites (cf, to L=50 that the authors use in this work). In fact one of the accents of 10.1073/pnas.1800589115 is on finite-size effects and comparison to the disordered case.

However, my main concern is not even with the originality of the work, but with the validity of its results. The authors use a very low bond dimension (xi<64, compared to xi=1000-2000, many time used in TEBD), and their method (1-site TDVP) doesn't allow them to estimate the discarded weight. Moreover, the comparison between the two shown bond-dimensions are done on a logarithmic scale and for different "disorder" realizations, which doesn't allow to estimate the time until which the results are reliable.

From Fig 1 it is obvious that this time is not longer than t~100 (which is also probably a conservative estimate). This can also seen from the comparison to TEBD, which the authors provide in their reply. Although the graph is also on a log-log scale, for W=4, the divergence with TEBD occurs already at t<100, where TEBD results appear to converge, as far as one ca n judge from this plot. While one can see that TEBD saturates the bond dimension 256 already at times t=20, the actually meaningful information is the accumulated discarded weight, which the authors don't provide.

As on can see from Fig 3 for t<100 and W=4, the drift in the dynamical exponent, which is one of the main results of the work is not significant, and in any case probably lies within the error bars (which are not shown for some reason). For W>5 the oscillations in the exponent do not allow to extract a coherent message, though the convergence issues I raised above, are presumably better there.

Another issue, which is not discussed in the paper is the convergence with respect to the time-step. Following a request by the first referee the authors do specify the time-step dt=0.1, in their reply, but don't provide any evidence that this time-step is sufficient to go to t=300 and W=8 for example.

In view of the above I agree with the report of the first referee that the conclusions of this work are most probably based on not converged data, and are therefore doubtful. I therefore do not think that the paper stands by the standards of SciPost.

Requested changes

See report

---

## Round 3 · Author Response

Dear Marcello Dalmonte,

In response to your invitation to do so, we would like to resubmit our manuscript "Many-body delocalization dynamics in long Aubry-André quasiperiodic chains." Based on the Referees' recommendations and comments, we have substantially reworked and expanded the manuscript. In particular, we have performed additional numerical calculations to address the concerns the referees have raised with respect to the reliability of our approach. The new version takes into account all recommendations and remarks of the Referees. Even though we disagree with some of these remarks, we have made changes in the manuscript in order to better explain the corresponding points. Our detailed responses to all comments by the Referees and the list of changes made are appended below.

We hope that the amended manuscript is to your approval.

Sincerely,
the authors
* * *
RESPONSE TO REPORT 1
* * *
REFEREE:
"The manuscript addresses many-body localization (MBL) in quasi-periodic on-site disorder in isotropic Heisenberg chain model - a paradigmatic model used for MBL studies. While the manuscript lists some of earlier works on this very topic [20,23,26] it misses, however, at least one early study PRB 87, 134202 (2013)."

OUR RESPONSE:
We thank the referee for pointing out this reference that is indeed relevant. We have included this reference (Ref. 15 of the revised version), which is now cited in the Introduction, Sec. 1. Furthermore, we have added a section Sec. 3.1.4 (see list of changes) that includes a detailed comparison with the existing literature, including the newly added Ref. 15.

REFEREE:
"Those earlier studies used typically exact diagonalization combined with the finite size scaling to estimate the transition to MBL. The present work considers experimentally accessible observables such as imbalance and distribution of spin densities whose time dynamics is evaluated using time-dependent variational principle with matrix product states. The authors report as their main finding (abstract and conclusion) the strong dependence of the system properties on the periodicity of the quasi-periodic potential which they trace to properties of the non-interacting problem. They stress that the decay of the anti-ferromagnetic order is faster than a power law (I believe on the delocalized side of the MBL transition)."

OUR RESPONSE:
The dependence on $\Phi$ is one of our main findings, as also stated in the abstract and in the conclusion. Another key finding is the weak dependence on system size of the critical disorder $W_c$. In the updated manuscript, we have substantially expanded on the latter finding, including a new figure (Fig. 3 of the updated manuscript). Furthermore, we observe indications that the decay is faster than power law.

REFEREE:
"The field of MBL is quite well established by now so making a significant contribution in the field is not as straightforward as say 5 years ago. In my opinion the work presented while providing some incremental value to the field, certainly does not meet the Scipost Physics criterion: "Articles provide in-depth, detailed reports of groundbreaking research within one or more subject areas." "

OUR RESPONSE:
We strongly disagree with this assessment. In our view, our study provides a very important addition to the existing literature by studying experimentally relevant observables in larger systems up to late times, which is of obvious relevance to current experiments. The questions that we address, including the accurate determination of the critical disorder, the dependence of its apparent location on the system size, the dependence on periodicity $\Phi$, and the character of decay in the ergodic phase are all of fundamental importance. The differences between the MBL transition in quasiperiodic and purely random systems is currently under active discussion and our work provides much needed numerical analysis to settle this debate. Indeed, while we were preparing the resubmission of this paper, more works appeared on the interacting AA model: 1902.07199, 1904.06928. We have included a Note Added where we now cite these references (Refs. 54 and 56 of the revised version).

REFEREE:
"The main claim, a strong dependence of the system dynamics on the periodicity of the quasi-periodic potential is a bit trivial and not new. It is explained more than 10 years ago in detail in V. Guearrera et al. New. J. Phys. 9, 107 (2007). In fact the on-site energy difference distribution (plotted in Fig.2 right panel of the manuscript) may be easily found analytically as proportional to $1/\sqrt{1−x^2}$ distribution with the amplitude of the effective disorder $W\sin(\pi \Phi)$ as it follows from that paper. Similar strong dependence on the quasi-periodicity was found in an interaction free problem by Major et al. Phys. Rev. A 98 053633 (2018). "

OUR RESPONSE:
First of all, we reiterate that a strong dependence on $\Phi$ is not the only "main claim" but rather one of key results of our work. We thank the referee for pointing out the important reference New. J. Phys. 9, 107 (2007), of which we were not aware. We have included it as Ref. 45 in the new version, and it is cited now in Sec. 3.1.2. The effective disorder mentioned in that work matches with our Fig. 2b, indeed as $W \sin(\pi \Phi)$.
The fact that the periodicity strongly affects the dynamics of the non-interacting problem was pointed out, as mentioned in our manuscript, also in Ref. [46], Phys. Rev. A 80, 021603 (2009) for the extended AA model. Our "main claim" is, however, not about the single-particle problem but rather about the influence of $\Phi$ on the MBL physics; this issue was not addressed in Ref. 45. Specifically, one of our key results is that the single-particle feature carries over to the many-problem directly and leads to a dependence of the dynamics and of the critical disorder of the MBL transition on the periodicity $\Phi$ - a non-trivial and to our knowledge new result. This should be contrasted to the non-interacting case where the critical disorder is always W=2 (in our units) for almost any irrational $\Phi$. That there are still interesting results from recent publications such as in [Phys. Rev. A 98 053633] even for the non-interacting problem underscores that there is still much more to learn in the interacting problem.

The fact that naive expectations based on properties of single-particle states in quasi-periodic systems may be insufficient and even misleading with respect to the MBL problem is additionally supported by a paper that has just appeared in SciPost: "Many-body localization in a quasiperiodic Fibonacci chain" by Nicolas Macé, Nicolas Laflorencie, Fabien Alet, SciPost Phys. 6, 050 (2019). Remarkably, these authors find that, contrary to naive expectations, the interaction does not enhance delocalization. One thus should study the many-body problem, which is what we do in our paper.

REFEREE:
"While the application of the variational principle MPS approach to time dynamics seems to be an original contribution, this has been already reported in [22] for a purely random case. So one can hardly give credit for that. "

OUR RESPONSE:
Indeed, in Ref. [22] we, together with collaborators, demonstrated that our method works well for the MBL problem with purely random disorder. This has allowed us to reliably study large systems and long times around the transition, including the ergodic side of the transition. In particular, we have demonstrated that the critical disorder for the disordered isotropic Heisenberg chain (the most standard model studied in the literature) is W_c \approx 5.5, which should be compared to the value 3.7 that was obtained in the most advanced exact-diagonalization study. Our work, which has attracted considerable attention in the community, thus demonstrated how important the finite-size effects on the MBL transition in random systems are.

Hence, it is natural and important to use the same computational approach also for the quasi-periodic case. This allows us to explore the physics of the MBL transition in the quasi-periodic case. Furthermore, using the same method is very favorable for comparing the results for random and quasi-periodic systems. We thus find it very strange that the fact that we used the TDVP approach which was used successfully for random systems is considered as an argument against our paper. We believe that it can only be viewed as an argument in favour of our work.

REFEREE:
"Especially, as time dependent dynamics using MPS (with e.g. TEBD) for MBL has been studied in a routine way, see [8], also
J. H. Bardarson, F. Pollmann, and J. E. Moore, Phys. Rev. Lett. 109, 017202 (2012)
P. Sierant, D. Delande, and J. Zakrzewski,Phys. Rev. A 95, 021601(R) (2017)
for different implementations.

The variational approach might have some advantage for long times due to the inherent energy conservation, the problem deserves a separate careful study in the MBL regime and close to it - comparisons as those in 1901.05824 do not suggest a clear advantage of the variational approach."

OUR RESPONSE:
We thank the referee for pointing out these references. We have included them as Refs. 30 and 31; they are cited in Sec. 1 of the revised version. Our method provides considerable advantages over TEBD specifically for the MBL problem (see also our response to the point 3 in the list of numbered remarks below). Indeed, an important advantage of the method is the inherent energy conservation, as opposed to the non-conservation of energy in e.g. TEBD. As a side note, in the language of 1901.05824 (which is Ref. 41 of our revised version) we use a 1TDVP algorithm, not a 2TDVP algorithm. The latter does not conserve energy for otherwise energy-conserving Hamiltonian dynamics due to the truncation step induced by the two-site approach. In fact, the authors of 1901.05824 argue that in general TDVP performs better than other MPS methods for various examples they consider. They recommend a hybrid approach, using the adaptive 2TDVP algorithm until a maximum bond dimension is reached, and then switching to 1TDVP. This approach is essentially very similar to our own, albeit somewhat more numerically efficient.

As suggested by the Referee, we have performed an analysis comparing the performance of 1TDVP and TEBD on the delocalized side of the MBL transition (W=4). It is presented now in the Appendix. These results confirm that 1TDVP performs substantially better, both from the point of view of convergence with the bond dimension and with respect to qualitative long-time behavior, see, in particular, Fig. 8. Specifically, (i) TDVP allows us to get convergence up to considerably longer time than TEBD; (ii) TEBD shows a spurious saturation of the imbalance at long time, which is apparently related to non-conservation of energy. With increasing bond dimension, this spurious feature is shifted towards longer times, and TEBD results move towards the correct behavior displayed by 1TDVP.

REFEREE:
"There is a strong believe that the system studied shows the mobility edge - the transition to MBL depends on energy (see Luitz et al Phys. Rev. B 91, 081103 (2015)). The present study considers a single initial anti-ferromagnetic state whose energy depends strongly on the disorder realization. This may affect the value of the critical disorder obtained. The question of the mobility edge is entirely ignored in the manuscript."

OUR RESPONSE:
A dependence of the critical disorder on energy in this system is indeed interesting, but we do not expect the essential properties of the MBL transition to depend on energy (as long as we are not too close to the ground state). A large majority of studies of MBL only target the middle of the spectrum and likewise ignore the question of a mobility edge. We thus leave the investigation of the energy-dependence of the critical point and verification of the universal character of the transition to future work.

REFEREE:
"Apart from these general remarks I believe that the authors should (in some future submission) consider the following remarks
1. Results for $\chi=32$ and $\chi=64$ seem unconverged (Fig. 1a and Fig.1b). Why not show convergent results for sufficiently large $\chi$?"

OUR RESPONSE:
In the updated manuscript, we have computed results up to $\chi = 128$, which are now shown in Fig. 4 (left panel), as well in Fig. 7 and 8 of the Appendix. In the left panel of Fig. 4 we now show the data for L=50 only up to the time t=180 for which a full convergence has been reached for $\chi=128$. We show also not-fully-converged results (i.e., for smaller $\xi$ or for longer times) in several figures since they represent an important information for the reader. Specifically, they allow to visulalize the convergence at moderately long times and a gradual loss of convergence at longer times, to see how the convergence time dependence on parameters, to compare the convergence in TDVP and TEBD, etc.

REFEREE:
"2. The present Fig.1 is unreadable - it requires huge enlarging to distinguish different curves. "

OUR RESPONSE:
Indeed, in each of the panels of Fig. 1 there are two curves that lay essentially on top of each other. In the left (W=4) and middle (W=5) panel this happens because the convergence with bond dimension has been reached, and we want to demonstrate this. In the right (W=8) panel these are two curves with lengths L=16 and L=50; here this demonstrates very weak length dependence. We have updated Fig. 1 to enhance clarity, by adding a zoomed region for each of the panels (see list of changes).

REFEREE:
"3. Let me mention that standard TEBD results with $\chi=250$ do not agree with those reported in Fig.1."

OUR RESPONSE:
Motivated by this and related comments, we have added a quite detailed comparison of TDVP and TEBD in the Appendix. The methods agree well at relatively short times $t \leq 50$, after which differences emerge, apparently due to non-conservation of energy of the TEBD method. As is seen in Fig. 7, and most clearly in Fig. 8, the $\chi = 256$ TEBD performs substantially worse than $\chi = 128$ TDVP with respect to convergence. Furthermore, TEBD shows a spurious saturation of the imbalance at longer times. On the other hand, with increasing $\chi$, this feature is shifted towards longer times, and TEBD data clearly move towards our TDVP results.

REFEREE:
"4. Why the authors do not provide information that would allow a reader to reproduce their results? What was the time step in time-dependent integration? What was the accuracy criterion (none - just $\chi$ was defined is one of the options)?"

OUR RESPONSE:
In the Appendix, we have now included more technical details, so that one can fully reproduce our results. In particular, we state the value of the time step; we have also included Fig. 9 (right panel) where we demonstrate that it was chosen small enough that the results are independent of it. As to the convergence with respect to the bond dimension, we have used the most reliable criterion -- calculated for various values of $\chi$ and checked the convergence.

In this connection, we would like to point out that we present much more technical details and have done more thorough analysis of convergence that most of DMRG papers studying related problems. As an example, the Referee has attracted our attention to the work P. (that the Referee apparently values high) Sierant, D. Delande, and J. Zakrzewski, Phys. Rev. A 95, 021601(R) (2017) which uses tDMRG. In that paper, we did not find any technical details, such as the bond dimension and the time step. Furthermore, the convergence criterion was not given; the data for different $\chi$ were not shown, so that the reader cannot judge on the quality of convergence etc. We would really appreciate a "fair play" on the side of the Referee with respect to our work.

REFEREE:
"5. The faster than power-law decay of the imbalance on the delocalized side, the main conclusion of the manuscript, is deduced from not converged results."

OUR RESPONSE:
Motivated by this remark of the Referee, as well as a related comment of the Second Referee, we have performed additional calculations for $W=4$ by using larger values of the bond dimension: 72, 96, and 128. Further, we have restricted the range of times for the corresponding curves in Fig. 4 (left panel) by t = 180, up to which the full convergence is reached for this value of $\chi$. The trend to increase of $\beta(t)$ is clearly there, although the absolute value of the increase in this range of times is indeed comparable to the error bars. We believe that this increase is a true physical property but we also agree that the data do not prove this unambiguously. We have correspondingly adjusted the wording in Sec. 3.2, as well in the Summary and in the abstract. We hope that future work (also stimulated by our paper) will manage to go in a controllable way to still longer times and to establish whether these deviations from power-law behaviour develop becomes strong and develop in a qualitatively different behavior at longer times.

REFEREE:
"6. The claim for the critical disorder to be Wc=5 is a rough estimate only. The text suggest error bar of the order of 0.5. Can the authors do better with the converged results?"

OUR RESPONSE:
We thank the Referee for this very useful question. It motivated us to substantially expand the analysis of the critical disorder, see Sec. 3.1.3 including a new Figure 3. Our analysis yields the critical disorder $W_c = 4.8 \pm 0.2$ for $\Phi = (\sqrt{5}-1)/2$ (modulo a standard "disclaimer" included in the paper: "Strictly speaking, the numerical analysis can never exclude a very slow delocalization that shows up only at times much longer than those achieved numerically. From this point of view, the numerically found values should be considered as lower bounds for $W_c$"). Moreover, we have added a detailed comparison with results on $W_c$ from the existing literature (mainly estimates based on exact diagonalization). Importantly, we find a very weak dependence of the apparent $W_c$ on system size. Consistently with this, previous estimates are in fair agreement with our result, underscoring that exact-diagonalization studies may perform rather well for determination of $W_c$ in quasi-periodic systems. This is in a contrast to the purely random case where we previously found much larger finite-size effects on $W_c$ and, correspondingly, considerably larger $W_c$ as compared to exact-diagonalization studies (5.5 vs 3.7).
This qualitative difference between the random and quasi-periodic systems is an important new result that sheds light on essential differences in the properties of the MBL phase in both models.

REFEREE:
"7. Authors claim a weak dependence on the system size for quasi-periodic potential in the introduction but they do not substantiate this claim further - showing just a single curve for L=16. How Wc depends on the system size?"

OUR RESPONSE:
We find no noticeable dependence of $W_c$ on system size between these two values (L=50 and L=16). A comparison is now shown in the new Figure 3. We do observe, however, a faster thermalization on the ergodic side for larger systems as evidenced by a larger value of the exponent $\gamma$.
* * *
RESPONSE TO REPORT 2
* * *
REFEREE:
"The authors numerically study the dynamics in a quasiperiodic (QP) interacting spin chain, which shows a MBL like transition as a function of the strength of the QP potential.

In their introduction the authors claim:

"The aim of the present work is to investigate the differences between purely random
and quasi-periodic disorder at system sizes inaccessible to exact diagonalization, using the
newly developed numerical technique of the time-dependent variational principle as applied
to matrix product states"

This statement is however misleading, since dynamics in QP interacting systems was studied using MPS based method in both [37] (L=100-200) and 10.1073/pnas.1800589115, for systems of up to 800 sites (cf, to L=50 that the authors use in this work). In fact one of the accents of 10.1073/pnas.1800589115 is on finite-size effects and comparison to the disordered case."

OUR RESPONSE:
Indeed, there have been other numerical studies of quasi-periodic systems using MPS, and we cite them in our paper. In particular, former Ref. 37 by Bar Lev et al is Ref. 32 of the new version; we have also included the PNAS paper by Znidaric and Ljubotina mentioned by the Referee, which is Ref. 34 of the revised version. However, these papers do not use TDVP which has essential differences with respect to other MPS methods (and important advantages over them for the present problem). Therefore, we see nothing misleading in our statement that we use the "newly developed numerical technique of the time-dependent variational principle...", especially since we use the split-step implementation of TDVP which was published in 2016.

As concerns the system size, one can, of course, go to larger L on the expense of shorter time t (for a given computational time). In our previous work on purely random system, Ref. 23, it was found that the finite-size effects are already quite small for L=50: the results for L=50 and L=100 were nearly identical. In view of this, we choose L=50 as the optimal length in this work. In the paper by Bar Lev et al. (Ref. 32 of the new version) quoted by the Referee, the t-DMRG approach was applied to the same quasiperiodic system but this was limited to rather short times (up to t=50 for W=3 in our notations). Presumably the discarded weight was kept below certain value in this study, although details are totally omitted. (We did not find any information on bond dimension, time step, convergence criterion, etc.) To explore deviations from the purely power-law behavior, one has to go to considerably longer times. It is also worth mentioning that the estimated position of the transition in this work, $W \approx 3$, is substantially below the value $W = 4.8 \pm 0.2$ that we find by using TDVP for considerably longer times around the transition.

The work by Znidaric and Ljubotina mentioned by the Referee (Ref. 34) focuses on the region deep in the ergodic regime, where their method (very different from ours) works well. On the hand, their approach becomes very inefficient when one moves closer to the MBL transition and is not appropriate for study of the vicinity of the transition.

Hence, our TDVP approach to quasi-periodic systems and the obtained results are new and complement the existing results in an important way.

As a response to this comment by the Referee, we have added a paragraph at the end of the Introduction discussing other MPS approaches and results in order to provide additional context to the reader.

REFEREE:
"However, my main concern is not even with the originality of the work, but with the validity of its results. The authors use a very low bond dimension (xi<64, compared to xi=1000-2000, many time used in TEBD), and their method (1-site TDVP) doesn't allow them to estimate the discarded weight. Moreover, the comparison between the two shown bond-dimensions are done on a logarithmic scale and for different "disorder" realizations, which doesn't allow to estimate the time until which the results are reliable."

OUR RESPONSE:
The issue of the bond dimension is similar to that of the system length. Of course, one can use the bond dimension 2000; however, this will go on the expense of the time t until which the dynamics is followed. One should thus find an optimal compromise. Close to the MBL transition, very large values of the bond dimension are less useful.

Motivated by this comment and a related comment by the First Referee, we have extended our calculations to larger bond dimensions, \chi = 96 and 128, for the case of disorder W=4. The corresponding results are shown in Fig. 4 (left panel), as well in Fig. 7 and 8. In the left panel of Fig. 4 we restricted results to the time t=180 for which a full convergence was reached for the largest values of the bond dimension. Furthermore, we have added a detailed Appendix (that includes Figs. 7, 8 and 9) with many additional benchmarks and an explanation of the key differences between TDVP and older MPS methods. We note also that a larger value of the bond dimension does not necessarily means better performance when different approaches are compared: as our results show, TDVP with $\chi = 128$ yields convergence for longer times that TEBD with $\chi= 256$.

We have also improved Fig. 1 to show zoomed regions for better clarity.

The results for the disorder average in the main text are presented intentionally, as we are interested in averaged transport properties. In response to the Referee's comment, we have included Fig. 9 (left panel) where we show data for a single disorder realization with different $\chi$. The time at which deviations start to be observed is essentially the same as when they are observed for averaged quantities for the same $\chi$.

REFEREE:
"From Fig 1 it is obvious that this time is not longer than $t \sim 100$ (which is also probably a conservative estimate). This can also seen from the comparison to TEBD, which the authors provide in their reply. Although the graph is also on a log-log scale, for $W=4$, the divergence with TEBD occurs already at $t<100$, where TEBD results appear to converge, as far as one can judge from this plot. While one can see that TEBD saturates the bond dimension 256 already at times $t=20$, the actually meaningful information is the accumulated discarded weight, which the authors don't provide."

OUR RESPONSE:
In the newly added Appendix, we have added a plot which shows the smallest value of the entanglement spectrum $\lambda_\chi$, a quantity closely related to the discarded weight that can also be compared to TDVP results. TDVP doesn't actually perform better than TEBD with higher bond dimension in terms of this quantity, which underscores that the application of TDVP is approximate and only numerically exact at short times. Convergence of this approximate application of TDVP can only be established by inspecting the convergence with bond dimension, akin to the results of Kloss et al. (Ref. 39 of the new version). In that paper, it was shown that TDVP can yield reasonable results for the disordered XXZ chain, essentially converged in bond dimension, at times far beyond the "numerically exact" convergence time. These findings were echoed by us in our study of the disordered XXZ chain; we now show that they hold also in the quasi-periodic case.

As for TEBD, for the largest value of the bond dimension that we used, $\chi=256$, is reached at a time $t=25$. For larger times, one can again study the convergence with $\chi$. This convergence is reached up to the time $t \approx 90$, which is well below the time $t=180$ up to which the convergence for TDVP with $\chi=128$ (see Figs. 4-left, 7, and 8). Up to $t \approx 90$, TDVP and TEBD agree. Beyond this time, where TEBD has not reached convergence, its results deviate from the converged results of TDVP.

REFEREE:
"As on can see from Fig 3 for t<100 and W=4, the drift in the dynamical exponent, which is one of the main results of the work is not significant, and in any case probably lies within the error bars (which are not shown for some reason). For W>5 the oscillations in the exponent do not allow to extract a coherent message, though the convergence issues I raised above, are presumably better there."

OUR RESPONSE:
Following the Referee's suggestion, we have added error bars to this plot (which is Fig. 4 of the revised version) obtained through a bootstrap procedure. Furthermore, motivated by this remark of the Referee, as well as a related comment of the First Referee, we have performed additional calculations for $W=4$ by using larger values of the bond dimension: 72, 96, and 128, in order to reach convergence up to longer times. Further, we have restricted the range of times for the corresponding curves in Fig. 4 (left panel) by t = 180, up to which the full convergence is reached for these values of $\chi$. The trend to increase of $\beta(t)$ is clearly there, although the absolute value of the increase in this range of times is indeed comparable to the error bars. We believe that this increase is a true physical property but we also agree that the data do not prove this unambiguously. We have correspondingly adjusted the wording in Sec. 3.2, as well in the Summary and in the abstract. We hope that future work (also stimulated by our paper) will manage to go in a controllable way to still longer times and to establish whether these deviations from power-law behaviour becomes stronger at longer times.

For W = 5 and larger (localized regime), our results (for $\Phi = (\sqrt{5}-1)/2$) converge quite well already with $\chi = 64$ for all considered times. The Referee asks about a "coherent message" related to these results. The first message is the slow oscillations on the localized side of the MBL that are absent in the purely random case. Second, we have added a plot, Fig. 3 (in response to a question raised by Referee 1), which shows the saturation of the imbalance as a function of $W$ for both $L=16$ and $L=50$. In this representation, the oscillations for e.g.\ $W = 5$ are washed out by averaging over a large time window, and vanishing of the imbalance decay within error bars is observed for $W \geq 5$. So, there is another "coherent message" based also on this data: the finite-size effects with respect to the critical disorder $W_c$
are very different in the purely random and quasi-periodic cases.

REFEREE:
"Another issue, which is not discussed in the paper is the convergence with respect to the time-step. Following a request by the first referee the authors do specify the time-step dt=0.1, in their reply, but don't provide any evidence that this time-step is sufficient to go to t=300 and W=8 for example."

OUR RESPONSE:
In the Appendix, we have added a plot showing convergence in time step, see right panel of Fig. 9. We have used parameters proposed by the Referee but this value of $\delta t$ is sufficient for other parameters as well. As this figure shows, a twice larger step $\delta t = 0.2$ is also fully sufficient.

In connection with this and some previous remarks of the Referee, we would like to point out the following. The Referee raises doubts in the validity of our work, criticizing us for not proving the convergence with the time step, using insufficient (from Referee's point of view bond dimension), etc. At the same time, the Referee empahsizes, in the very beginning of the report, an important contribution made by the paper by Bar Lev et al (Ref. 32 of the new version). However, this latter paper does not provide any technical details (information on bond dimension, time step, convergence criterion, etc.) and yields a critical value W_c that is too low by as much as 60%, so that it would be completely disqualified if the Referee would apply the same criteria. We would really appreciate fairness on the side of the Referee with respect to our work.

---

## Round 3 · List of Changes

SUMMARY OF CHANGES
* * *
We have substantially revised the paper in order to take into account recommendations and remarks by both Referees. The changes are listed below:

1) In response to a comment by Referee 2, we have added a paragraph in the Introduction discussing the relation of our method to other MPS-based approaches.

2) In response to comments of both Referees, we have expanded Fig. 1, showing insets for greater clarity.

3) Following a recommendation of the Referee 1, we have substantially expanded the analysis of the critical disorder, which is now presented in Sec. 3.1.3. In particular, we have added a new Figure 3 showing imbalance decay as a function of the disorder strength $W$ for system sizes $L = 16$ and 50. This figure visualizes smallness of finite-size effects and allows us to extract the critical disorder $W_c = 4.8 \pm 0.2$, which is now emphasized in abstract and in conclusion.

4) In connection with this, we have added Sec. 3.1.4 with a comparison to results from the existing literature.

5) Since we have substantially expanded Sec. 3.1 in the part of the analysis of critical disorder and comparison to existing literature, we have introduced and additional level of structure of Sec. 3.1 and correspondingly added subheadings (Sec. 3.1.1 to Sec. 3.1.4). This will help the reader to see the logical structure of Sec. 3.1.

6) In response to comments by both Referees, we have added in the left panel of Fig. 4 showing the "flowing power law coefficient" $\beta(t)$ for $W = 4$ new results for higher bond dimension $\chi = 72$, 96, and 128 (for the length $L=50$). Furthermore, we have restricted the curves by the time $t=180$, up to which the full convergence has been reached with these values of $\chi$.

7) Following the recommendation of Referee 2, we have added error bars in Fig. 4.

8) In response to comments of the Referees, we have modified the wording of our conclusion concerning the increase of $\beta(t)$ in Sec. 3.2, in Conclusion, and in the abstract.

9) Following recommendations of the Referees to provide more technical details and in response to the questions concerning the comparison of TDVP with other MPS approaches, we have added a detailed Appendix. There, we explain the numerical method, provide technical details and benchmarks and compare to a different MPS-based numerical approach (time-evolved block decimation = TEBD). The Appendix contains newly added figures 7, 8 and 9.

10) We have added the references suggested by the referees to the reference list and cited them appropriately in the text.
We have also included a few more references, including those required for the added discussion in the appendix.

11) In the very end of the main body of the paper, we have added a Note concerning two recently appeared preprints on related subjects.

12) In addition, we have done various small changes in formulations and have fixed detected typos.

---

## Editorial Decision

editor-in-charge_assigned